# Interrogating ligand-receptor interactions using highly sensitive cellular biosensors

Maximilian A. Funk [1], Judith Leitner [1] ✉, Marlene C. Gerner [2], Jasmin M. Hammerler[2], Benjamin Salzer [3,4], Manfred Lehner [3,4], Claire Battin[1], Simon Gumpelmair[1], Karin Stiasny [5], Katharina Grabmeier-Pfistershammer[6] & Peter Steinberger [1] ✉

Interactions of membrane-resident proteins are important targets for therapeutic interventions but most methods to study them are either costly, laborious or fail to reflect the physiologic interaction of membrane resident proteins in trans. Here we describe highly sensitive cellular biosensors as a tool to study receptor-ligand pairs. They consist of fluorescent reporter cells that express chimeric receptors harboring ectodomains of cell surface molecules and intracellular signaling domains. We show that a broad range of molecules can be integrated into this platform and we demonstrate its applicability to highly relevant research areas, including the characterization of immune checkpoints and the probing of cells for the presence of receptors or ligands. The platform is suitable to evaluate the interactions of viral proteins with host receptors and to test for neutralization capability of drugs or biological samples. Our results indicate that cellular biosensors have broad utility as a tool to study protein-interactions.

In multicellular organisms the interactions of membrane-resident proteins play essential roles in immune responses but also in numerous other biological processes. Consequently, they are a primary area of research and important targets for therapeutic interventions. Viruses also rely on the interaction of their attachment proteins with surface receptors to enter their host cells[1]. Protective humoral immunity as well as recombinant antibodies and small molecules that block such interactions can confer effective protection of the host organism.

Numerous methodologies have been developed to study receptor-ligand binding. Many rely on recombinant proteins representing the ectodomains of the binding partners. They can be studied in protein-protein interaction assays such as ELISA or surface plasmon resonance (SPR) in which either receptor or ligands are immobilized on a stationary phase. Alternatively labelled recombinant proteins representing the extracellular domains of receptor or ligands can be

probed with cells expressing their cognate interaction partners. However, the interaction of receptors with soluble ligands in the fluid phase does not reflect the two-dimensional interaction of cell-resident receptor-ligand pairs in trans[2,3]. Assays that study such receptor-ligand interactions under conditions where both molecules are presented in their natural conformation on the cell surface in the context of a lipid bilayer represent a more physiological setting. These assays mirror the local concentrations as well as the constraints put upon the interaction partners. Cell conjugation assays can be used to measure the interaction of distinct pairs of receptors and ligands expressed on fluorescently labelled cells[4–6]. However such assays require fairly strong interactions and are difficult to standardize since cell-cell interactions are influenced by a plethora of factors. The adhesion frequency assay and the thermal fluctuation assay have been devised to measure molecular interactions across two opposing cell membranes[2,7]. Cell-

[1]Center for Pathophysiology, Infectiology and Immunology, Institute of Immunology, Division for Immune Receptors and T cell activation, Medical University of Vienna, Vienna, Austria. [2]Division of Biomedical Science, University of Applied Sciences FH Campus Wien, Vienna, Austria. [3]St. Anna Children's Cancer Research Institute, Vienna, Austria. [4]Christian Doppler Laboratory for Next Generation CAR T Cells, Vienna, Austria. [5]Center for Virology, Medical University of Vienna, Vienna, Austria. [6]Department of Dermatology, Medical University of Vienna, Vienna, Austria. ✉e-mail: judith.a.leitner@meduniwien.ac.at; peter.steinberger@meduniwien.ac.at

membrane mimicking platforms such as supported lipid bilayers can also be used to detect and track the interaction of membrane-bound counterreceptors at the single molecule level when used in conjunction with high-end microscopy. All of these, however, are highly sophisticated methodologies that require considerable expertise and special equipment and are therefore unfit for broad applications such as routine diagnostic testing.

Cells engineered to carry reporter genes were shown to have utility as biosensors to measure the presence of a wide variety of natural and synthetic ligands[8–12]. However, such systems lack versatility since their use is limited to receptor-ligand interactions that generate signals that induce reporter activation. In the pioneering work by Irving and Weiss the intracellular signaling domain of one receptor is fused to the ligand-binding ectodomain of another receptor[13]. Importantly, ligand binding as well as signaling capability of the donor molecules was retained. This principle has been used in chimeric antigen receptors that use single-chain antibody fragments binding surface antigens to efficiently redirect T cells and other effector cell populations towards tumor target cells[14,15]. Of note, chimeric receptors harboring ectodomains derived from receptors such as CD4 or desmoglein were also demonstrated to efficiently retarget T cells[16–18]. Altogether this indicates that in chimeric receptors extracellular recognition can be linked to intracellular signaling pathways of choice. Consequently, chimeric receptors harboring potent signaling domains could be combined with compatible reporter cells to generate highly sensitive cellular biosensors for the interrogation of receptor-ligand interactions. They should have utility to detect the presence of cognate binding partners on adjacent cells with great sensitivity as well as to assess the capability of inhibitors such as blocking antibodies to interfere with receptor engagement.

Here we have explored this hypothesis and have endowed various cell-resident receptors and ligands with intracellular signaling domains and expressed them in highly sensitive fluorescent reporter cells. We demonstrate that chimeric molecules harboring the ectodomains of the immune checkpoints PD-1 or PD-L1 are capable of mediating strong reporter activation upon engagement and can be used to evaluate immune checkpoint inhibitors targeting the PD-1 – PD-L1 axis. We show that cellular biosensors function with a broad range of engineered cell surface molecules including type II transmembrane proteins. Finally, we have generated cellular biosensors based on signaling-competent chimeric virus entry receptors harboring the ectodomains of CD46, CD4 and ACE2. The presence of cells expressing the respective virus entry proteins measles virus hemagglutinin, HIV-1 gp160 and SARS-CoV-2 spike protein induced potent reporter activation. This signal could be effectively and dose-dependently disrupted by neutralizing antibodies. Our results indicate that cellular biosensors expressing chimeric virus receptors can be deployed in a type of diagnostic surrogate neutralization assay that avoids the use of pseudo-typed viruses or costly recombinant proteins. Furthermore, the assay can be applied to characterize the effectiveness of drugs blocking cell surface receptor-ligand interactions such as checkpoint inhibitors or anti-viral antibodies.

## Results

### Cellular biosensors as a highly versatile tool to analyze receptor-ligand interaction

We hypothesized that chimeric receptors consisting of a receptor ectodomain combined with an intracellular T cell activation domain expressed on highly sensitive reporter cell lines could be applied to study the interaction with their respective cell-expressed ligands. Recently, suitable reporter cell lines such as Jurkat E6-1 NFκB-eGFP or Jurkat E6-1 Triple parameter (NFκB-eCFP; NFAT-eGFP; AP-1-mCherry) reporters (JE6-1-TPR) were developed in our laboratory[19,20]. We therefore created a chimeric molecule, where the ectodomain of human PD-1 was fused to a CD28-transmembrane domain and a CD3ζ signaling

domain (PD-1-ζ). Upon ligand-engagement the T cell receptor activation pathway downstream of CD3ζ is activated and ultimately results in the nuclear translocation of NF-κB (Fig. 1a). Lentiviral transduction of PD-1-ζ resulted in high expression levels on JE6-1 NFκB-eGFP reporter cells (Fig. 1b). Co-culture of these biosensor cells with cells expressing PD-L1 resulted in eGFP expression in the reporter cells (Fig. 1c and Supplementary Fig. 1a, b). Antibodies targeting PD-L1, such as Avelumab and Atezolizumab are used as immune checkpoint inhibitors to increase anti-tumor immunity in cancer patients[21]. Addition of these antibodies to the co-cultures potently disrupted PD-1-ζ engagement and led to a dose-dependent reduction of eGFP expression, while there was no such effect with a matching isotype control antibody (Fig. 1d, e and Supplementary Fig. 1c, d). We also investigated if the PD1-ζ biosensor cells could be used to test the potency of small molecule inhibitors of the PD1/PD-L1 interaction. Therefore, the compounds INCB[22], I3[23], BMS1[24] and BMS202[25,26] were analyzed in the PD1-ζ biosensor assay. INCB proved to be more effective than I3 (IC$_{50}$ 2.3 nM vs. 162 nM) (Supplementary Fig. 1e–g). BMS1 and BMS202 however did not show significant inhibitory effect even at very high concentration (10 μM) and exerted cytotoxic effects when used in higher concentrations (Supplementary Fig. 1h–k). This unexpected result underscores the importance of testing small molecule inhibitors in functional cellular assays.

We also generated a chimeric molecule based on PD-L1, the major PD-1 ligand and confirmed strong expression of the PD-L1-ζ molecule in biosensor cells (Fig. 1f, g). Co-culture with cells expressing PD-1 resulted in strong induction of eGFP while the PD-1 directed immune checkpoint inhibitor Nivolumab but not the isotype control reduced PD-1 mediated activation of PD-L1-ζ biosensor cells (Fig. 1h and Supplementary Fig. 2a–c). Besides PD-1, CD80 has also been identified as an interaction partner for PD-L1[27]. However, recent work did not detect enhanced conjugation between cells expressing PD-L1 and CD80 indicating that CD80/PD-L1 interaction mainly occurs in cis on the same cell surface but not in trans between cells. We performed interaction assays of PD-L1-ζ biosensor cells with cells expressing high levels of CD80 or with cells expressing CD86, which is not an interactor for PD-L1. Co-culture with cells expressing CD80 but not with cells expressing CD86 induced low-level activation of PD-L1-ζ biosensor cells indicating that CD80 is able to engage PD-L1 in trans to some degree (Fig. 1i and Supplementary Fig. 2d). Finally, we explored if PD-L1-ζ biosensors could be triggered by a PD-L1 antibody bound to BW cells by the Fc-receptor mFcγRIIB (mCD32B) (Supplementary Fig. 2e). Indeed, cell-bound PD-L1 antibody induced a potent biosensor signal, establishing a universal and independent method to validate biosensor functionality (Supplementary Fig. 2 f, g). Taken together, our results indicate that molecules that primarily function as ligands can also be used for the creation of cellular biosensors and that immune checkpoint inhibitors as well as the interaction of putative binding partners can be efficiently evaluated in our system.

### Integration of type II and viral cell surface molecule ectodomains into cellular biosensors

PD-1 and PD-L1 like the CD3-ζ-chain are typical type I transmembrane proteins with a C-terminal intracellular part and are therefore likely to retain functionality in a chimeric molecule. Therefore, we next addressed whether type II transmembrane proteins can also be integrated into our biosensor system. After a start codon the sequence of CD3ζ was fused to the N-terminal intracellular part of the type II surface protein 4-1BBL (Fig. 2a). The expression of the 4-1BBL-ζ construct in JE6.1 TPR cells was similar to wildtype 4-1BBL (Fig. 2b). An interaction assay with stimulator cells expressing 4-1BB showed signaling competence of the 4-1BBL-ζ molecule (Fig. 2c–e). The signal intensity correlated dose-dependently to the amount of stimulator cells placed in co-culture with the reporter cells. As few as 2500 stimulator cells (reporter-to-stimulator ratio, 20:1) still induced a clearly measurable

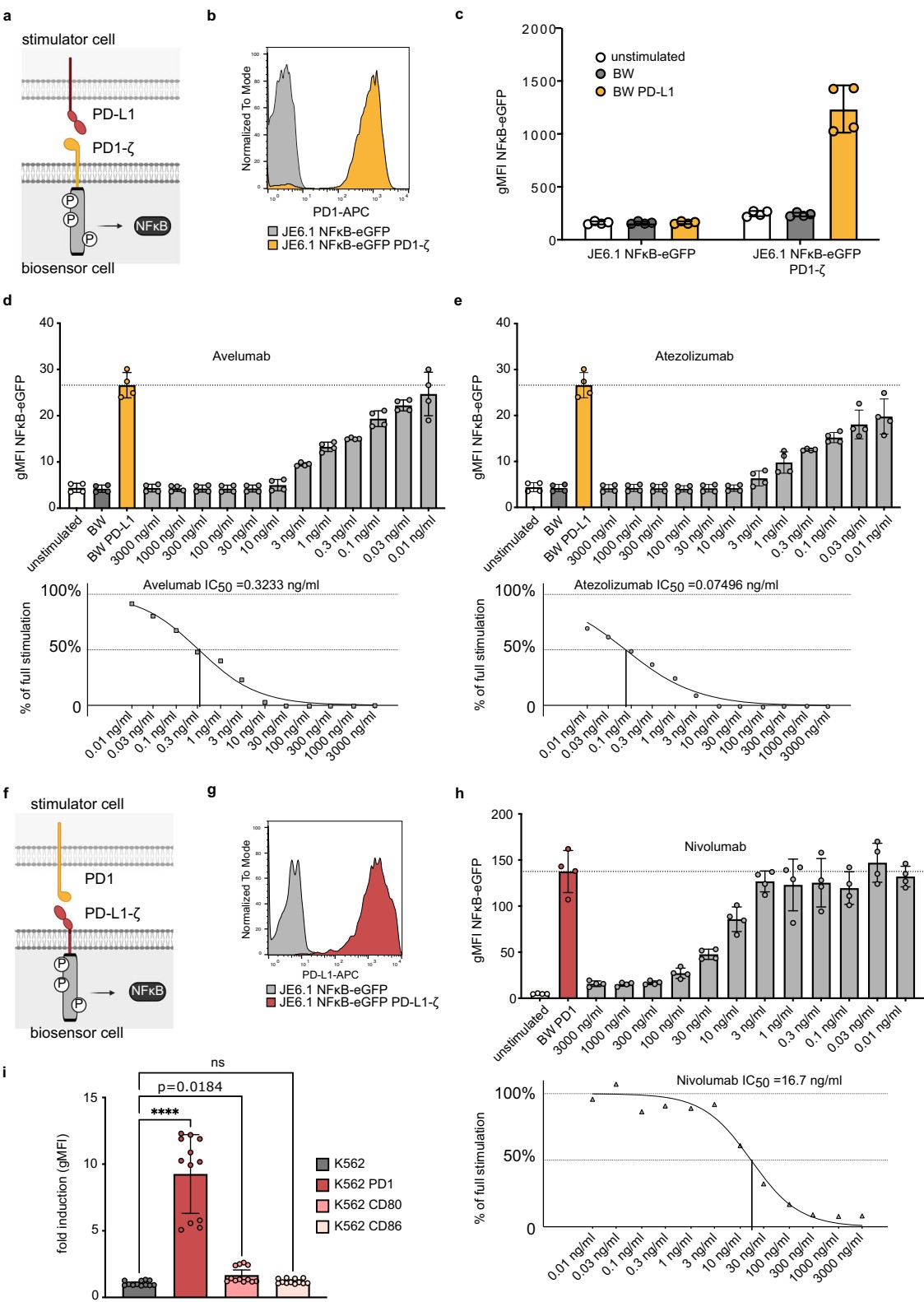

signal underlining the high sensitivity of the cellular biosensors (Fig. 2f). 4-1BB, the counterreceptor for 4-1BBL is an important target for immunotherapeutic approaches and the agonistic 4-1BB antibodies utomilumab and urelumab are being investigated to boost T cell activation in the setting of various cancer entities[28,29]. We used 4-1BBL-ζ expressing biosensor cells to test whether these antibodies block 4-1BB/4-1BBL interaction. Polyclonal 4-1BB-directed antibodies served as positive control. While there was some blockade by the polyclonal

antibodies, we did not detect any signal attenuation with either monoclonal antibody indicating unimpaired interaction between 4-1BB and 4-1BBL in the presence of both antibodies (Fig. 2g). For Urelumab this is in accordance to previous studies that reported no interference with the 4-1BB/4-1BBL complex[30]. Utomilumab is supposed to block 4-1BBL binding to 4-1BB in a competitive manner, albeit with very low overlap of the binding sites. Therefore, high concentrations of utomilumab are necessary to achieve even a modest

**Fig. 1 | Cellular biosensors are a highly versatile tool to analyze receptor-ligand interaction. a** Scheme depicting the interaction of PD-L1 on stimulator cell with the chimeric PD-1-ζ receptor on biosensor cells. **b** Expression of PD-1-ζ receptor on JE6.1 NFκB-eGFP reporter cells. **c** JE6.1 NFκB-eGFP and JE6.1 NFκB-eGFP PD-1-ζ were co-cultured as indicated. (*n* = 2 experiments, performed in duplicates). Data is presented as individual replicates with mean ± SD of geometric mean fluorescence intensity (gMFI) NFκB-eGFP. BW is short for BW5147 a murine thymoma cell line. **d**, **e** Effect of PD-L1 blocking antibodies Avelumab (**d**) and Atezolizumab (**e**) on the stimulation of JE6.1 NFκB-eGFP PD-1-ζ biosensor cells. (*n* = 2 experiments, performed in duplicates). Top panels show gMFI values, data is presented as individual replicates with mean ± SD of gMFI NFκB-eGFP. Bottom panels show normalized data and non-linear regression curve fitting. IC50 values are indicated. **f** Scheme depicting the interaction of PD-1 on stimulator cells with the chimeric PD-L1-ζ receptor on biosensor cells. **g** Expression of PD-L1-ζ on JE6.1 NFκB-eGFP reporter cells. **h** Effect of the PD-1 blocking antibody nivolumab on the stimulation of JE6.1 NFκB-eGFP PD-L1-ζ biosensor cells. (*n* = 2 experiments, performed in duplicates). Data presented analogous to (**d**, **e**). **i** JE6.1 NFκB-eGFP PD-L1-ζ biosensor cells were co-cultured with stimulator cells as indicated. (*n* = 3 experiments, 4 replicates each). Data is presented as individual replicates with mean ± SD. For statistical analysis Kruskal-Wallis test for unpaired, non-normally distributed data with Dunn's multiple comparisons test was performed. (**** *p* < 0.0001; ns, not significant). Source data for this figure are provided as a Source Data file.

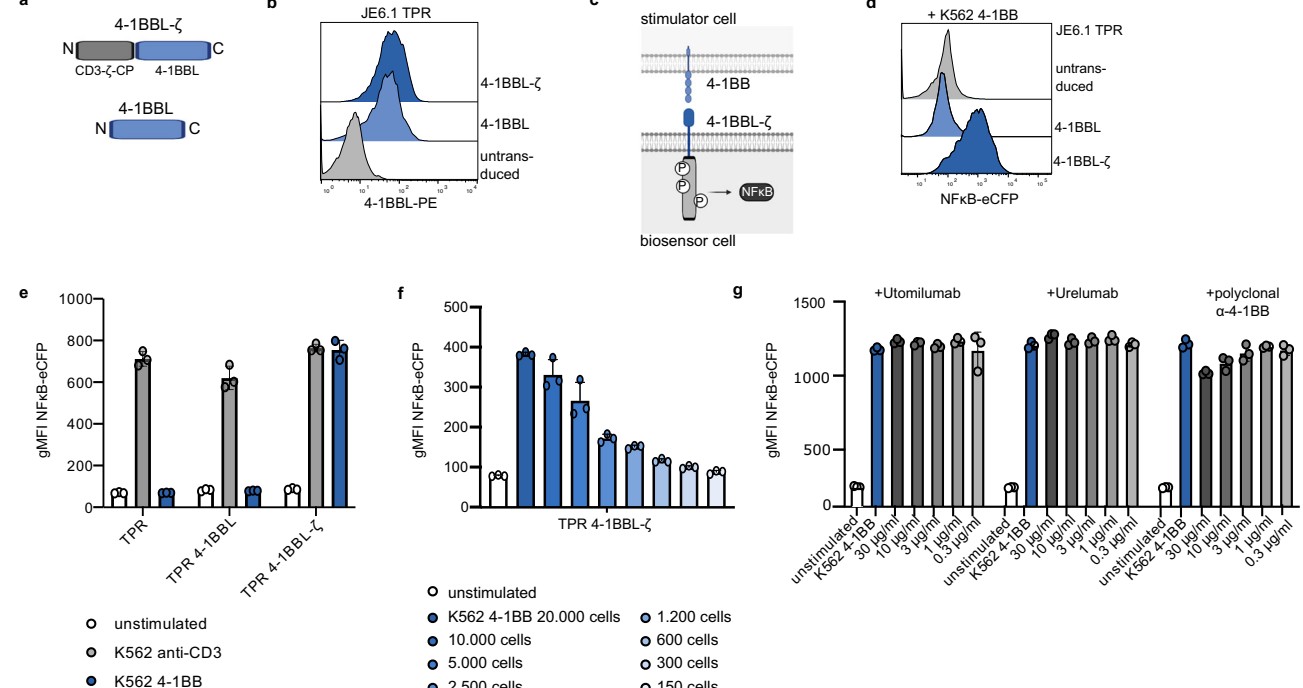

**Fig. 2 | Integration of type II molecule ectodomains into cellular biosensors. a** Scheme depicting 4-1BBL-ζ and wildtype 4-1BBL constructs. **b** JE6.1 TPR cells or JE6.1 TPR cells expressing 4-1BBL or 4-1BBL-ζ were stained with a 4-1BBL antibody and assessed by flow cytometry. **c** Scheme depicting interaction of 4-1BBL-ζ on biosensor cells with 4-1BB on stimulator cells. **d**, **e** JE6.1 TPR, JE6.1 TPR 4-1BBL and JE6.1 TPR 4-1BBL-ζ were co-cultured with K562 cells expressing a membrane-bound anti-CD3 antibody fragment or K562 cells expressing 4-1BB and analyzed for reporter gene expression. **d** Representative histograms showing reporter gene expression of biosensor cells stimulated with K562 4-1BB is shown (data presented as NFκB-eCFP fluorescent intensity (FI)) **e** Representative experiment, performed in triplicates. Data is presented with mean ± SD of gMFI NFκB-eCFP. Data of one repeat experiment is provided within Source Data file. **f** 5×10⁴ JE6.1 TPR 4-1BBL-ζ cells were co-cultured with indicated count of K562-41BB cells (representative experiment, performed in triplicates). Data is presented with mean ± SD of gMFI NFκB-eCFP. Data of one repeat experiment is provided within Source Data file. **g** JE6.1 TPR 4-1BBL-ζ biosensor cells were co-cultured with stimulator cells as indicated. K562 4-1BB stimulators were preincubated with the indicated concentration of 4-1BB antibodies (*n* = 1 experiment, performed in triplicates). Data is presented as individual replicates with means ± SD of gMFI NFκB-eCFP. Source data for this figure are provided as a Source Data file.

reduction of binding of 4-1BBL protein to 4-1BB[30,31]. Our data indicate that utomilumab is inefficient in blocking the interaction of cell-expressed 4-1BB and 4-1BBL in trans.

Investigating the interaction of cell surface molecules derived from pathogens with proteins expressed on the membrane of host cells is another application for our biosensor system. The genome of the human cytomegalovirus (CMV) encodes several putative transmembrane proteins. One of these proteins, UL11, has been shown to interact with immune cells and was subsequently identified as the first viral ligand for CD45[32]. We generated a construct encoding a UL11-chimera containing a C-terminal ζ-signaling domain and an N-terminal Strep-II-tag sequence for detection (Supplementary Fig. 3a). The construct was expressed in JE6.1 TPR cells where *PTPRC* (encoding CD45) was knocked out to avoid engagement of the UL11-ζ biosensor by reporter cell-expressed CD45 (Supplementary Fig. 3b). Co-culture with

CD45 expressing K562 wildtype (wt) cells but not with *PTPRC*^KO K562 cells induced the activation of UL11-ζ biosensor cells (Supplementary Fig. 3c–e). These results show that the biosensor cells are suitable to study interactions of various types of cell surface receptors, including type II transmembrane and viral molecules, with their ligands.

**Probing complex samples for the presence of interactors**
One potential use of our system is to screen complex cellular samples for the presence of binding partners to orphan ligands and receptors. In proof of principle experiments we probed activated PBMC samples with biosensors expressing PD-1-ζ, PD-L1-ζ or 4-1BBL-ζ (Fig. 3a). We used *TRAC/TRBC*^KO-JE6.1-Nur77-mKO2 reporter cells, which are not activated by anti-CD3/anti-CD28-beads and unlike JE6.1 NFκB-reporter cells do not respond to TNFα produced upon activation of PBMCs. The expression of these biosensors on these reporter cells was confirmed

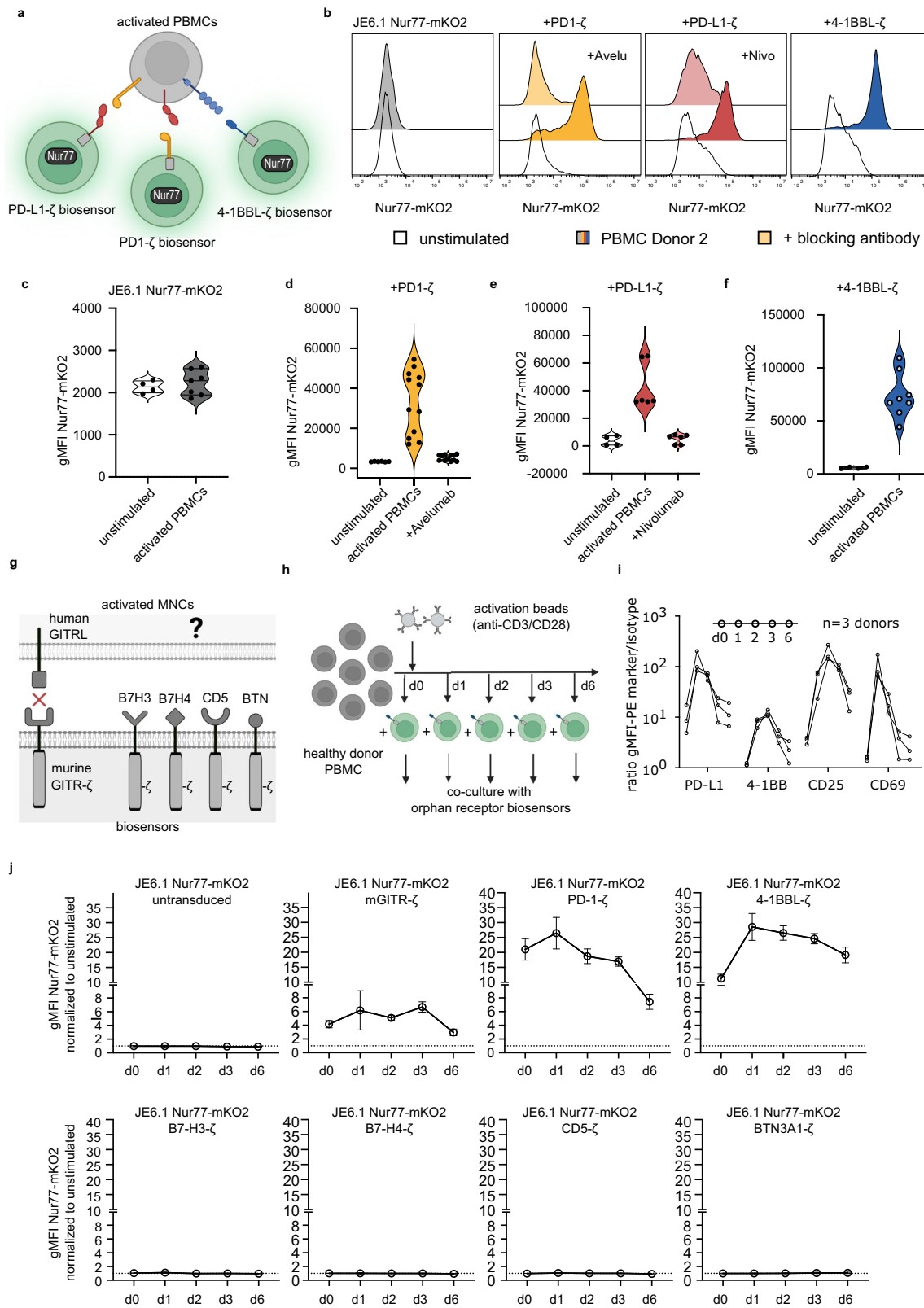

by flow cytometry (Supplementary Fig. 4a). Anti-CD3/anti-CD28 stimulated PBMCs expressed high levels of PD-1, PD-L1 and 4-1BB (Supplementary Fig. 4b). Biosensor cells expressing PD-1-ζ, PD-L1-ζ or 4-1BBL-ζ were strongly activated by stimulated PBMCs (Fig. 3b–f). The activation mediated by PD-1-ζ and PD-L1-ζ was inhibited in the presence of antibodies to PD-1 and PD-L1 confirming specific activation of

the cellular sensors by the respective binding partners (Fig. 3b, d, e). Based on these encouraging results, we then generated biosensor constructs representing four orphan ligands and receptors namely B7-H3, B7-H4, CD5 and BTN3A1 (Butyrophilin 3A1) (Fig. 3g). B7-H3, B7-H4 and BTN3A1, which are structurally related to the B7 family, have been described as inhibitory ligands that dampen T cell responses by

**Fig. 3 | Probing complex samples for the presence of binding partners.**
**a** Scheme depicting the interaction of anti-CD3/anti-CD28 activated PBMCs with different biosensor cells. **b–f** Activated PBMCs were co-cultured with control JE6.1 Nur77-mKO.2 reporter cells (expressing no ζ-construct), or reporter cells expressing PD1-ζ, PD-L1-ζ or 41BBL-ζ as indicated. In some conditions, the blocking antibodies avelumab and nivolumab were added to the respective biosensor reporter cells (JE6.1 Nur77-mKO2 PD1-ζ and PD-L1-ζ). **b** Histograms showing reporter gene expression of one representative donor. **c–f** Pooled data of independent co-culture assays is shown as violin plots with individual replicates (unstimulated condition performed in duplicates for every experiment, control n = 2 experiments, 3 donors, 1-4 replicates/donor, PD1-ζ n = 3 experiments, 3 donors, 2-6 replicates/donors, PD-L1-ζ n = 2 experiments, 2 donors, 2-4 replicates/donor, 4-1BBL-ζ n = 2 experiments, 3 donors, 2-4 replicates/donor. **c** JE6.1 Nur77-mKO2 **d** JE6.1 Nur77-mKO2 PD1-ζ **e** JE6.1

Nur77-mKO2 PD-L1-ζ **f** JE6.1 Nur77-mKO2 4-1BBL-ζ. **g** Scheme depicting the concept of the orphan ligand detection assay. JE6.1 Nur77-mKO2 mGITR-ζ serve as negative control as mGITR has been reported to interact with hGITR-L on activated PBMCs[39]. **h** Scheme depicting assay setup. Healthy donor PBMCs were activated using anti-CD3/anti-CD28 coated beads and placed in co-culture with biosensor reporter cells. Co-cultures were sampled at the indicated time points to analyze biosensor reporter cell activation. **i** Expression kinetic of PD-L1, 4-1BB, CD25 and CD69 on activated PBMCs (n = 3 donors) over 6 days. **j** Graphs show reporter gene induction of untransduced reporter cells and the indicated biosensor cells at the indicated time points. Data is presented normalized to unstimulated biosensor reporter cells (mean with 95% CI). (n = 3 donors, 1-3 repeats for each donor at each time point). Raw gMFI values and source data for this figure are provided as a Source Data file.

interacting with as of yet unidentified receptors[33–35]. Of note this function is distinct from the involvement of BTN3A1 in the activation of Vγ9Vδ2 TCRs, which is dependent on the intracellular B30.2 domain that is not present in the BTN3A1-ζ molecule used in our study[36]. CD5 is genetically and structurally linked to CD6 but unlike CD6, no ligands have been unequivocally described for CD5. Proteins representing the extracellular domain of CD5 bind to immune cells indicating the presence of as of yet unidentified ligands for CD5[37,38]. The constructs were expressed in JE6.1-Nur77-mKO2 reporter cells and their expression was confirmed by flow cytometry (Supplementary Fig. 4c). Specific antibodies in conjunction with mCD32B expressing cells were used to demonstrate that they mediated strong reporter activation upon engagement (Supplementary Fig. 4d–g). We then performed co-culture experiments with the JE6.1-Nur77-mKO2 reporter cells expressing B7H3-ζ B7-H4-ζ, CD5-ζ and BTN3A1-ζ to assess whether these biosensors would detect the presence of unknown ligands in resting or activated PBMCs derived from different donors (Fig. 3h). Strong activation of PBMCs was confirmed by staining for activation markers (Fig. 3i). Biosensors expressing PD1-ζ and 4-1BBL-ζ were used as positive controls. As negative control we used untransduced JE6.1-Nur77-mKO2 reporter cells and reporter cells expressing a biosensor that was expected to not interact with surface molecules expressed on human PBMCs. We choose murine (m)GITR-ζ for this purpose, since it was reported to not ligate to the human orthologue of murine GITR-L[39]. In these experiments we found no reactivity of any of our "orphan biosensors" with resting and activated human PBMCs, whereas biosensors expressing PD1-ζ and 4-1BBL-ζ were strongly activated by all tested PBMC samples (Fig. 3j). Unexpectedly, we detected a modest signal with the of JE6.1-Nur77-mKO2 mGITR-ζ biosensor cells in all samples (Fig. 3j and Supplementary Fig. 4h). We thus re-visited a potential interaction of murine GITR with human GITR-L and probed JE6.1 Nur77-mKO2 mGITR-ζ with stimulator cells expressing high levels of murine and human GITR-L (Supplementary Fig. 5a–c). We found that these cells induced activation of these reporter cells albeit to much weaker extend than cells expressing murine GITR-L. The activation was also reversible by addition of a human GITR-L directed antibody (Supplementary Fig. 5d–e). Accordingly, this antibody attenuated mGITR-ζ signaling induced by activated PBMCs (Supplementary Fig. 5f). This corroborates both - an interaction between mGITR and human GITR-L and the high sensitivity of our biosensor assay since this interaction was not detected when probing cells expressing high levels of mGITR with hGITR-L fusion proteins in an earlier study[39].

## Cellular biosensors to study measles virus receptor engagement
We hypothesized that the assay could be used to study virus-host interactions by co-culturing stimulator cells expressing viral entry proteins and biosensor cells expressing the respective cellular-receptor-CD3ζ chimeric construct. CD46 is a major entry receptor for the measles virus. Virion binding to CD46-expressing host cells is mediated by the interaction with the measles virus glycoprotein hemagglutinin[40]. We created K562 cells expressing a C-terminally V5-

tagged measles hemagglutinin (K562 hemagglutinin). A chimeric receptor consisting of the CD46 ectodomain fused to CD3ζ was expressed in JE6.1 TPR biosensor cells lacking endogenous *CD46* (JE6-1-TPR-*CD46*^KO) (Fig. 4a–c). Co-culture of K562 hemagglutinin with CD46-ζ biosensor cells but not reporter cells expressing the signaling deficient CD46-ctrl molecule induced a strong fluorescent signal (Fig. 4d–f). A previous study by Vongpunsawad and colleagues has characterized hemagglutinin binding to its cellular receptors CD46 and CD150 and has identified important amino acid residues that are involved in binding to the respective receptors. Mutation of these amino acids resulted in selectively receptor-blind hemagglutinin proteins (ΔCD46 and ΔCD150)[41]. We expressed the respective hemagglutinin proteins on K562 cells and co-cultured them with CD46-ζ biosensor cells. While the wildtype and ΔCD150 hemagglutinin proteins both efficiently activated CD46-ζ biosensor cells, K562 cells expressing the ΔCD46 receptor-blind variant of hemagglutinin induced a significantly reduced activating signal, corroborating the previous results (Fig. 4g, h). These results also demonstrate the exquisite sensitivity of the assay, as the ΔCD46 hemagglutinin retains a weak capacity to interact with CD46, which was not shown by Vongpunsawad et al. Importantly, preincubation of K562 hemagglutinin with vaccinated donor serum also significantly reduced the activation of CD46-ζ-biosensor cells, indicating that this assay may be useful to detect neutralizing antibodies against this virus (Fig. 4g, h). This hypothesis is further supported by the fact that the addition of intravenous immunoglobulin (IVIG) preparations induced a concentration-dependent inhibition of the CD46-ζ biosensors in co-culture with K562 hemagglutinin (Fig. 4i). These results suggest that the biosensor cell assay can be applied to study virus-host receptor interactions and is sensitive to molecule-intrinsic reduced affinity or extrinsic influences, such as blocking antibodies.

## Characterizing broadly neutralizing antibodies against HIV-1
To this date, HIV is a major health concern and numerous attempts have been undertaken to develop therapeutic neutralizing antibodies[42]. We presumed that applying the biosensor assay, a platform could be created that allows for quick characterization of the efficacy of such neutralizing antibody drugs. We created chimeric molecules harboring the CD4 ectodomain fused to the ζ-chain alone or to a 4-1BB-ζ signaling domain (CD4-1BBζ) for increased biosensor signal intensity and expressed CD4-BBζ in JE6.1 TPR CD4^KO cells (Supplementary Fig. 6a–b). A construct encoding the BaL-HIV-gp160 precursor protein was expressed in K562 cells to yield cells harboring the gp120-gp41-complexes on their surface (K562 gp160) (Fig. 5a, b). Co-culture of CD4-41BBζ biosensor cells with K562 gp160 cells induced weak biosensor activation (Supplementary Fig. 6c–e). The signal could be further enhanced by co-expressing CD86 on gp160^+ cells. (Supplementary Fig. 6f) K562 cells co-expressing CD86 with HLA DR1 or HLA DR10 could be used to detect the low-affinity interaction of the CD4 ectodomain with MHC class II molecules (Supplementary Fig. 6g, h). To further improve the readout of the CD4-gp160

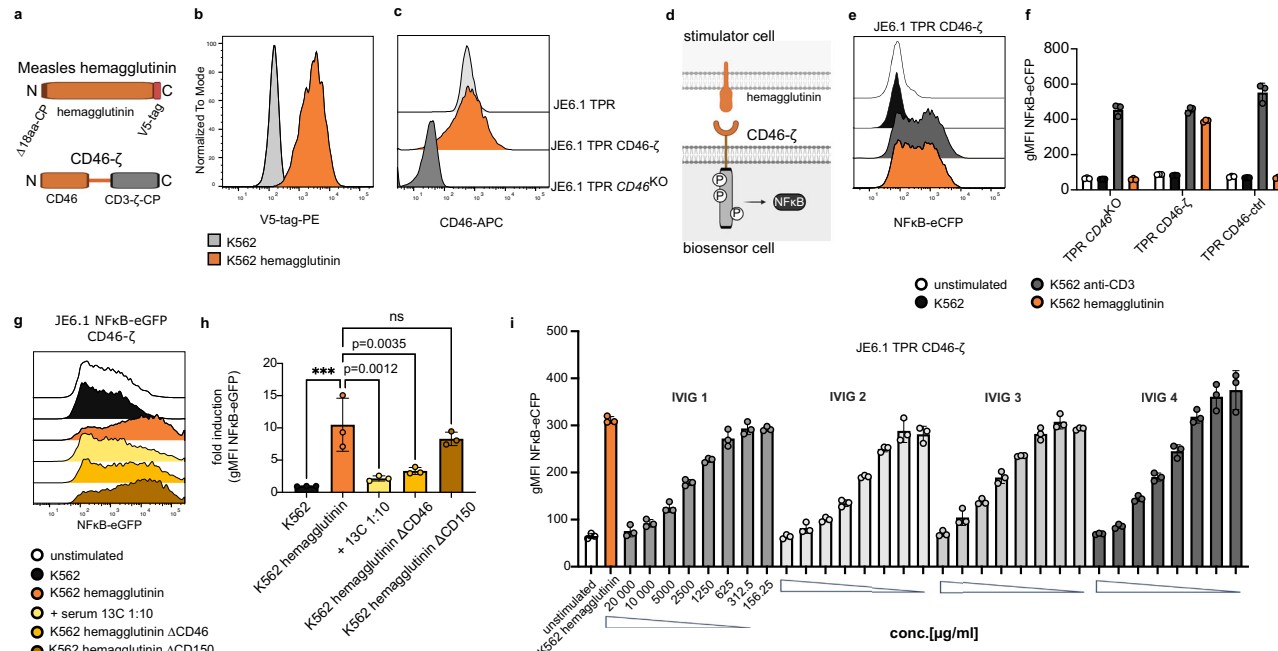

**Fig. 4 | Cellular biosensors to study measles virus receptor engagement.**
**a** Scheme depicting measles hemagglutinin with Δ18aa cytoplasmic domain and C-terminal V5-tag modifications and the CD46-ζ chimeric receptor. **b** Expression of measles hemagglutinin on K562 RFP stimulator cells. **c** Expression of CD46 on JE6.1 TPR, JE6.1 TPR-*CD46*KO and JE6.1 TPR-*CD46*KO CD46-ζ. **d** Scheme depicting the interaction of measles hemagglutinin on stimulator cell with the CD46-ζ chimeric receptor on biosensor cells. **e, f** JE6.1 TPR *CD46*KO, JE6.1 TPR CD46-ζ and CD46-ctrl were co-cultured with the indicated cell lines **e** Representative histograms showing reporter gene expression of JE6.1 TPR CD46-ζ biosensor cells stimulated with indicated stimulator cells (data presented as NFκB-eCFP FI). **f** One experiment was performed in triplicates. Data is presented as replicates with mean ± SD of gMFI NFκB-eCFP. **g, h** JE6.1 NFκB-eGFP CD46-ζ biosensor cells were co-cultured with stimulator cells as indicated. For one condition K562-hemagglutinin cells were preincubated with vaccinated donor serum at a dilution of 1:10. **g** Representative

histograms showing reporter gene expression of JE6.1 NFκB-eGFP CD46-ζ biosensor cells stimulated with indicated stimulator cells (data presented as NFκB-eGFP FI). **h** Graph shows pooled data from independent experiments ($n = 3$, each performed in triplicates). The triplicate means of NFκB-eGFP gMFI values were normalized to unstimulated cells. Data is presented as mean fold induction values of the three experiments with mean ± SD. For statistical analysis ordinary one-way ANOVA with Dunnett's multiple comparisons test for unpaired, normally distributed data was performed (***$p = 0.0004$; ns $p > 0.05$). **i** JE6.1 TPR CD46-ζ biosensor cells were co-cultured with K562 hemagglutinin stimulator cells. The stimulators were preincubated with IVIG preparations at the indicated concentrations ($n = 1$ experiment, performed in triplicates). Data is presented as individual values with mean ± SD of gMFI NFκB-eCFP. Source data for this figure are provided as a Source Data file.

interaction assay we expressed the CD4-ζ molecule in JE6.1 Nur77-mKO2 reporter cells that are highly sensitive to CD3ζ mediated T cell activation (Fig. 5c, d). The CD4-ζ Nur77-mKO2 reporter cells were validated by co-culturing them with K562, K562 gp160 or BW mCD32B alone or pre-incubated with a CD4 antibody. As expected, K562 gp160 and BW mCD32B preincubated with the CD4 antibody induced strong expression of the mKO2 reporter protein (Fig. 5e, f). As hypothesized, pre-incubation of K562 gp160 cells with well-characterized HIV-neutralizing antibody drugs resulted in dose-dependent inhibition of CD4-ζ biosensor activation (Fig. 5g). Titration of the different antibodies indicated that the CD4 antibody SIM.2 and the anti-gp120 antibody NIH45-46-G54W had the highest potency among the agents tested in our study. These results indicate that the biosensor assay can be applied to study the effect of therapeutic, neutralizing antibodies and could serve as easily manageable and cost-effective high-throughput platform for the development of such therapeutics.

### Development of a highly sensitive surrogate neutralization test for SARS-CoV-2

Since the beginning of 2020, the SARS-CoV-2 pandemic has had major impacts on the health of millions of people worldwide[43]. The SARS-CoV-2 spike glycoprotein binds the cellular receptor ACE2 to enter host cells[44]. Antibodies that block this interaction are generally thought to have a primary role in the protection that is conferred upon SARS-CoV-2 infection or vaccination. Due to the still incompletely understood mechanisms of host immunity to the virus there is still a great need for

tools to study antibody responses to SARS-CoV-2. We therefore set out to establish the biosensor-based interaction assay as diagnostic surrogate neutralization test (Fig. 6a). A wildtype SARS-CoV-2-spike protein with an N-terminal Strep-II-tag was expressed in K562 stimulator cells. For stable surface expression in the pre-fusion conformation, the spike protein was modified by deleting the arginine amino acids at the furin-cleavage site (RRRdel) and introducing the K1043P and V1044P mutations (Fig. 6b)[45,46]. The ACE2-ζ was furnished with an N-terminal c-myc tag for detection (Fig. 6b). Both constructs achieved high expression in K562 stimulator and JE6.1 NFκB-eGFP reporter cells, respectively (Fig. 6c, d). The signaling capability of the ACE2-ζ biosensors was evaluated microscopically. ACE2-ζ biosensor cells were cultured with K562 cells expressing the SARS-CoV-2 spike protein. A neutralizing serum was added to the coculture at the indicated dilution steps. When used at a dilution of 1:10 this serum completely abrogated the signal whereas eGFP expression gradually returned with higher dilutions of neutralizing serum (Fig. 6e). Sera collected from one donor prior to and two weeks after the first and second vaccination with ChAdOX1 were tested in our ACE2-ζ biosensor assay. Again, the ACE2-ζ construct induced a strong signal (Fig. 6f). After the second dose the serum showed clear, titratable neutralization capability, indicating seroconversion (Fig. 6g). The inhibition at each serum dilution step was expressed as percent of ACE2-ζ reporter activation in absence of serum. Non-linear regression curve fitting then allowed to calculate IC$_{50}$ values for each serum sample. To compare the biosensor-based surrogate neutralization assay to established assays we tested a cohort of 181 donors and calculated IC$_{50}$

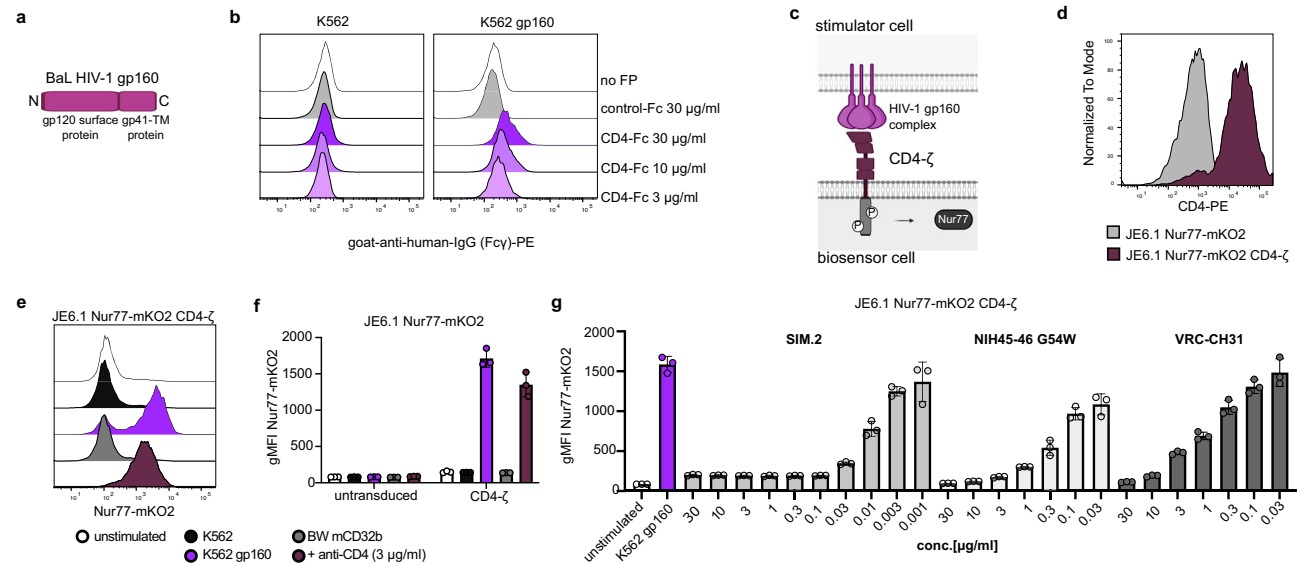

**Fig. 5 | Characterizing broadly neutralizing antibodies against HIV-1. a** Scheme depicting the BaL HIV-1 gp160 construct **b** Binding of CD4-Fc fusion protein to K562 cells (left panel) and K562 gp160 cells (right panel). **c** Scheme depicting interaction of gp160 complex on stimulator cell with CD4-ζ chimeric receptor on JE6.1 Nur77-mKO2 biosensor cells. **d** Expression of CD4 on JE6.1 Nur77-mKO2 and JE6.1 Nur77-mKO2 CD4-ζ. **e, f** JE6.1 Nur77 reporter cells and JE6.1 Nur77 CD4-ζ biosensor cells were co-cultured with stimulator cells as indicated (*n* = 4 experiments, performed in triplicates). **e** Representative histograms showing reporter gene expression of JE6.1 Nur77-mKO2 CD4-ζ biosensor cells cultured with indicated stimulator cells

(data presented as Nur77-mKO2 FI) **f** Representative experiment, performed in triplicates. Data is presented as individual replicates with mean ± SD of gMFI Nur77-mKO2. Data of three repeat experiments is provided within Source Data file. **g** JE6.1 Nur77 CD4-ζ biosensor cells were co-cultured with K562 gp160 stimulator cells. The stimulators were preincubated with neutralizing antibodies as indicated (*n* = 2 experiments, performed in triplicates). Data of representative experiment is presented as replicate values with mean ± SD of gMFI Nur77-mKO2. Data of one repeat experiment and source data for this figure are provided in the Source Data file.

values for those serum samples that showed reactivity at a dilution of at least 1:3. The cohort was previously evaluated with the electrochemiluminescence immunoassay Elecsys® Anti-SARS-CoV-2 that measures antibodies to the RBD of the SARS-CoV-2 spike protein. Overall, the biosensor assay reached a sensitivity of 100 % and a specificity of 97 % when compared to the electrochemiluminescence immunoassay. $IC_{50}$ values of 77 donors that had reactivity in both assays were correlated to the cut-of-index (COI) determined by the electrochemiluminescence immunoassay. Since values were not normally distributed, we performed Spearman correlation and found a strong negative correlation (r -0.6940, $p < 0.0001$) indicating lower $IC_{50}$ values at higher antibody concentrations (Fig. 6h). The gold-standard for the determination of serum-neutralizing capacity is the live-virus neutralization test (NT)[47]. 12 samples were therefore further characterized with such a NT. Figure 6i shows strong negative correlation (r -0.8094, $p = 0.0021$) of the sample $IC_{50}$ values with NT titers, indicating lower $IC_{50}$ values correlating to higher NT-titers. Finally, to exclude an impact of the stabilizing mutations on our SARS-CoV-2 biosensor neutralization assay, we performed an investigation with unmutated spike. For this, SARS-CoV-2 spike was transiently expressed in HEK293 cells. Untransfected HEK and HEK transiently expressing B7-H3 served as negative controls (Supplementary Fig. 7a). In co-culture, biosensor cells could be identified by staining for CD28 (Supplementary Fig. 7b). HEK293 cells expressing the unmutated SARS-CoV-2 spike but not untransfected or control-transfected HEK293 cells induced a detectable signal in ACE2-ζ biosensors. This signal was fully reversible by addition of neutralizing serum. (Supplementary Fig. 7c, d). Altogether this indicates that the biosensor-based interaction assay has potential to be used as surrogate test for the detection of SARS-CoV-2 neutralizing antibodies.

## Cellular biosensors to assess the neutralization of SARS-CoV-2 variants

As SARS-CoV-2 evolved novel variants with distinct features have appeared and asserted dominance over previous viral strains. These newer variants of concern show higher transmissibility and reduced susceptibility to host immune mechanisms[48,49]. Therefore, there is a great need to quickly identify immune escape variants and determine the capacity of sera and therapeutic antibodies to neutralize novel SARS-CoV-2 variants. The biosensor-based assay is potentially well-suited to quickly assess emerging variants of concern since it does neither depend on the time-consuming and costly generation recombinant proteins nor on the establishment of pseudo-typed viruses harboring spike proteins of concern. We created stimulator cells expressing the spike proteins of the wildtype strain and the B.1.617.2, AY.4.2 (Delta) and B.1.1.529 BA.1, BA.2 and BA.5 (Omicron) strains at high levels (Fig. 7a). All variants could induce ACE2-ζ biosensor activation in a similar range (Fig. 7b, c). The antibodies Regdanvimab, Tixagevimab, Cilgavimab and the combination of the latter two (Evusheld) were evaluated to determine, if the biosensor assay is suitable to assess neutralization capacity of therapeutic and prophylactic antibodies (Fig. 7d–g). Regdanvimab showed great efficacy in blocking the wildtype strain spike protein and retained reduced but detectable efficacy against both Delta strains B.1.617.2 and AY.4.2. However, Omicron variants B.1.1.529 BA.1, BA.2 and BA.5 were not neutralized by Regdanvimab which is in accordance with the current clinical recommendation to refrain from using Regdanvimab since the appearance of the Omicron variant (Fig. 7d)[50]. Tixagevimab also showed a severe loss of activity against Omicron variants BA.1 and BA.2 and a complete loss of activity against the BA.5 variant while Cilgavimab retained activity in the Omicron spectrum. As expected Evusheld showed additional neutralization capacity against the wild type and Delta variants when compared to Tixagevimab or Cilgavimab alone, while the activity against Omicron variants closely matched that of Cilgavimab (Fig. 7e–g). This is also in line with previous reports[51–53].

We also analyzed wildtype spike-reactive IVIG preparations regarding their capacity to block the activation of ACE2-ζ biosensors by variant spike proteins. Generally, both IVIG preparations showed higher $IC_{50}$ values when compared to monoclonal antibodies under

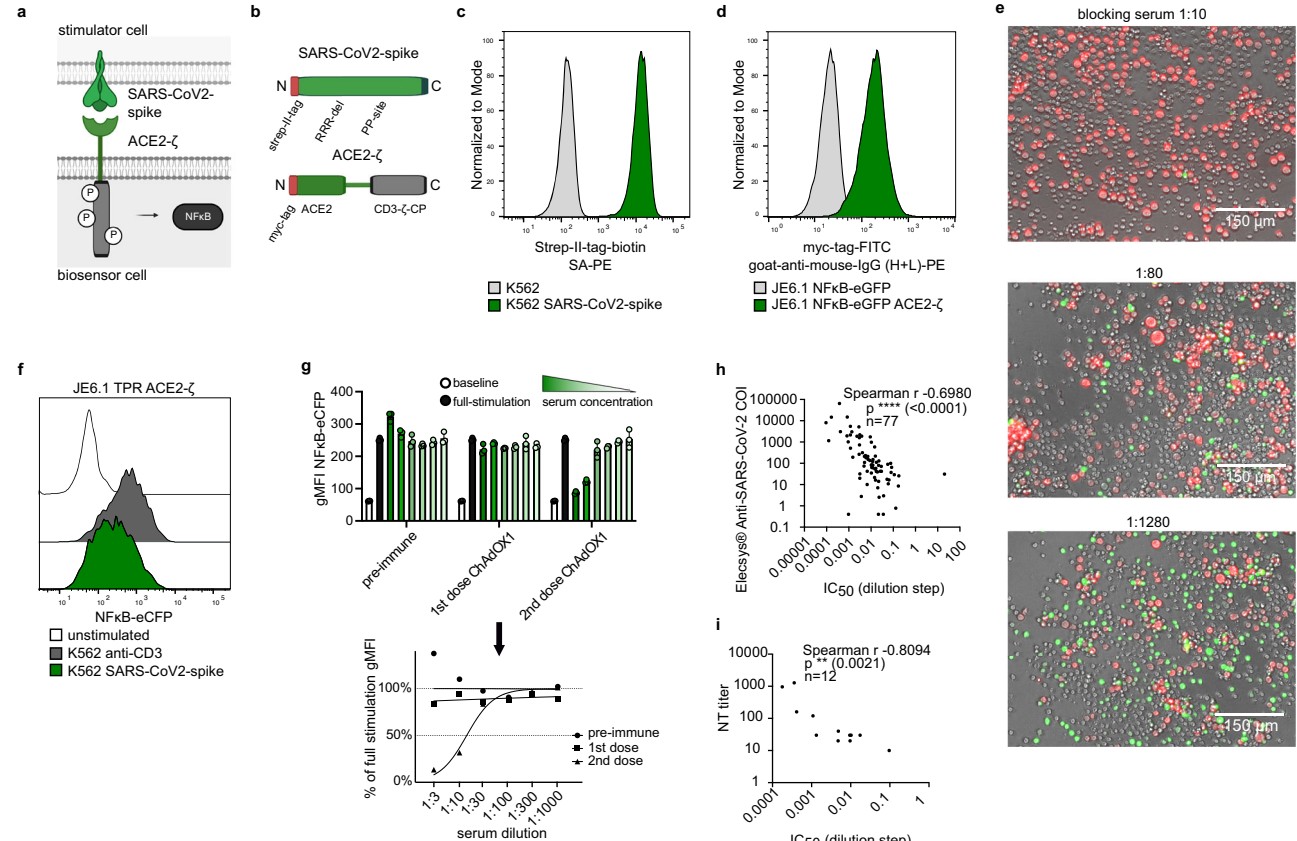

**Fig. 6 | Development of a highly sensitive surrogate neutralization test for SARS-CoV-2. a** Scheme depicting interaction of SARS-CoV-2 spike on stimulator cell with ACE2-ζ chimeric receptor on biosensor cells. **b** Scheme depicting SARS-CoV-2 spike with N-terminal Strep-II-tag modification and stabilizing mutations (top) and ACE2-ζ chimeric receptor with N-terminal myc-tag (bottom). **c** Expression of SARS-CoV-2 spike on K562 RFP stimulator cells. **d** Expression of ACE2-ζ on JE6.1 NFκB-eGFP biosensor cells. **e** Fluorescence microscopy images representative of three replicates. K562 SARS-CoV-2 spike CD86 cells expressed RFP (red). JE6.1 NFκB-eGFP ACE2-ζ cells expressed eGFP upon activation (green). Stimulator cells were incubated with vaccinated donor serum at the indicated dilutions. Corresponding gMFI-values assessed by flow cytometry are provided in the Source Data file. **f, g** JE6.1 TPR ACE2-ζ biosensor cells were co-cultured with K562 SARS-CoV-2 spike stimulator cells. **f** Representative histograms showing reporter gene expression of JE6.1 TPR ACE2-ζ biosensor cells cultured with indicated stimulator cells (data presented as NFκB-eCFP FI, $n = 1$ experiment). **g** Stimulator cells were preincubated with sera obtained from a donor prior to and after vaccination with ChAdOX1. Each condition was performed in triplicates. Data is presented as individual values with mean ± SD of gMFI NFκB-eCFP (top). Each mean was then normalized to full stimulation without serum and non-linear regression curve fitting was performed. For the blocking serum an $IC_{50}$ value was calculated (bottom). **h** A cohort of 77 serum donors with reactivity to SARS-CoV-2 determined by an electroluminescence immunoassay was evaluated with the biosensor surrogate neutralization test. $IC_{50}$ values were then correlated to cut-off index (COI) values of the electroluminescence immunoassay (higher COI indicate higher antibody levels). Spearman correlation for non-normally distributed samples was performed (**** $p < 0.0001$, two-tailed). **i** A cohort of 12 serum donors was evaluated by a live-virus NT. $IC_{50}$ values determined by the biosensor surrogate neutralization test were correlated to the NT titers. Spearman correlation for non-normally distributed samples was performed (** $p = 0.0021$, two-tailed). Source data for this figure are provided as a Source Data file.

most conditions. Interestingly however, while for both preparations reactivity against the Omicron variants was weakest, none had completely lost their neutralizing capability to a given variant (Fig. 7h, i). This could be explained by the mixture of broad polyclonal antibodies from multiple donors that are present in IVIG preparations. This data may imply that while monoclonal antibodies are highly effective against a specific variant, in times of rapidly emerging new variants with immune escape features high titer IVIG preparation could be the preferred drugs to prevent severe disease in patients at risk. We also tested individual serum samples to determine if the test was suitable to diagnose an individual's immune response to the different variants (Supplementary Fig. 8a–c). Two vaccinated donors and a vaccinated and convalescent donor were tested against all variants. All donors showed reactivity to the variants. However, the clear immune escape pattern of the Omicron variants was not as evidently reflected on the individual level as it was when assessing IVIG preparations. As the samples were acquired in late 2022, previous exposure may have influenced antibody levels in these donors. These results highlight that

there may be considerable differences in an individual's susceptibility to infection with a given variant. Overall, we show that the biosensor interaction assay can be applied to study the immune response to different strains of SARS-CoV-2 and to validate the efficacy of monoclonal antibodies and IVIG preparations as therapeutic or prophylactic drugs.

## Discussion

Chimeric receptors have been extensively used to redirect T cells or NK cells but also other immune cells towards malignant cells[54]. Here we introduced chimeric receptors harboring ectodomains of cell surface molecules fused to signaling domains into reporter cells to generate cellular biosensors for receptor-ligand interactions. Importantly, we show that this system is versatile and permissive for the integration of ectodomains derived from different types of receptors. We generated cellular biosensors based on the primary immune checkpoint PD-1 and also on its major ligand PD-L1 exemplifying that also ligands can be converted in chimeric receptors and used in this system. Our results

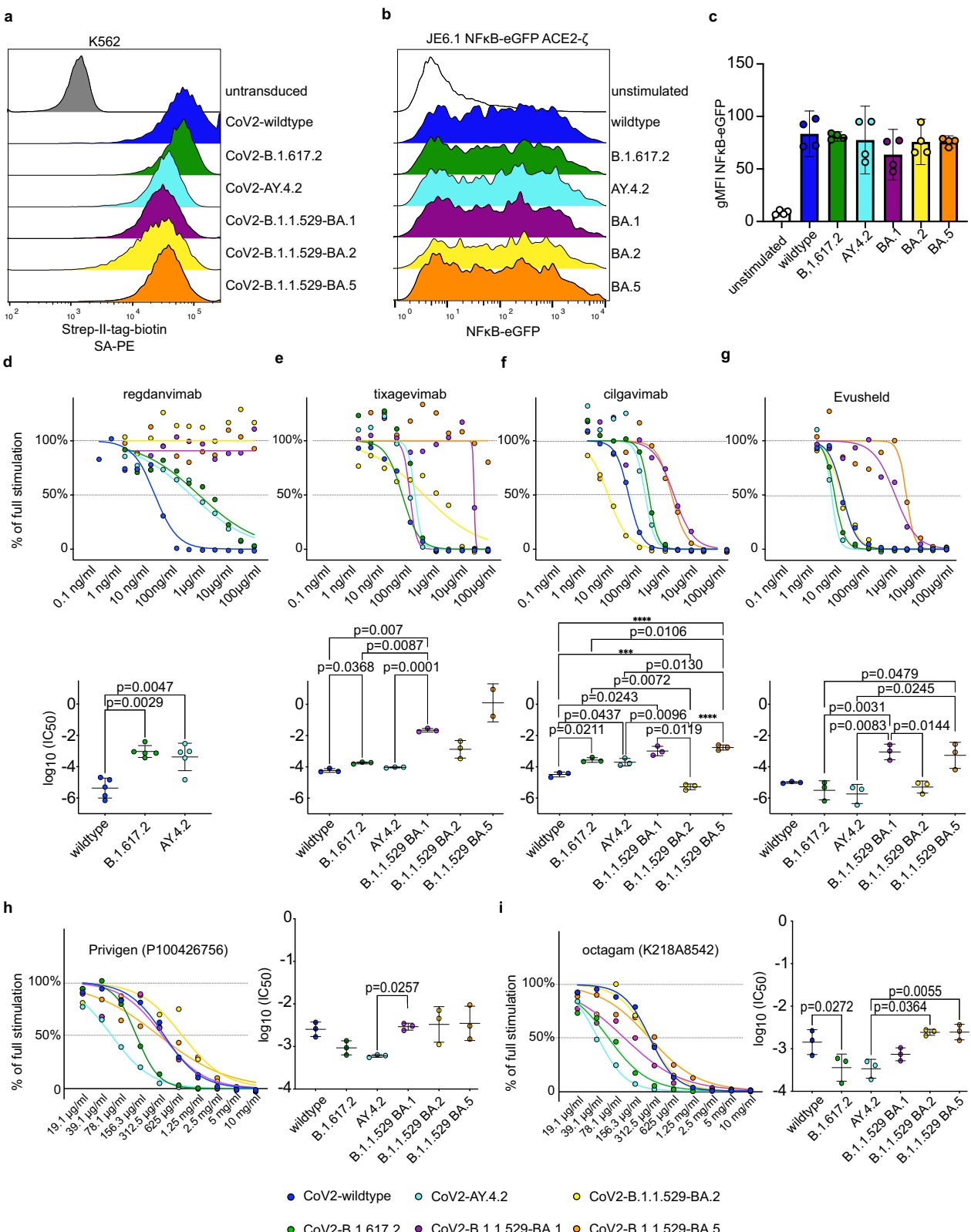

indicate that cellular biosensors have utility to evaluate immune checkpoint inhibitors. The IC$_{50}$ values obtained for clinically used PD-1 and PD-L1 antibodies were somewhat lower compared to those obtained in a functional reporter assay that detects blockade of PD-1 mediated inhibition[55]. This could be owed to the fact that in the previous assay the interaction of the reporter cells with stimulator cells is stronger since it also involves interaction of the TCR/CD3 complex with

membrane-bound anti-CD3 scFv displayed on the stimulator cells. Furthermore, we observed that PD-L1-ζ biosensors were weakly stimulated in presence of CD80-expressing cells indicating that cell-resident CD80 and PD-L1 molecules can interact in trans. Chaudhri et al. could not detect an interaction between CD80 and PD-L1 in trans, which is most likely due to a lower sensitivity of the cell conjugation assay used in their study[4]. We also show the generation of sensitive cellular sensors

**Fig. 7 | Cellular biosensors to assess the neutralization of SARS-CoV-2 variants.** **a** Expression of indicated Strep-II-tagged SARS-CoV-2 spike variants on K562 RFP stimulator cells as assessed by flow cytometry. **b**, **c** JE6.1 NFκB-eGFP ACE2-ζ biosensor cells were co-cultured with K562 RFP stimulator cells expressing different SARS-CoV-2 spike proteins. **b** Representative histograms showing reporter gene expression of JE6.1 NFκB-eGFP ACE2-ζ biosensor cells cultured with indicated stimulator cells (data presented as NFκB-eGFP FI). **c** Pooled data is presented as replicate values with mean ± SD of gMFI NFκB-eGFP (*n* = 2 experiments, performed in duplicates). **d**–**g** Stimulator cells were preincubated with neutralizing antibodies at the indicated concentrations. (Evusheld represents a combination of Tixagevimab + Cilgavimab) (**d** *n* = 5 experiments, **e**–**g** *n* = 3 experiments, performed in

duplicates or triplicates). Replicate means were normalized to full stimulation with the respective variant spike without antibody. Top panels show a representative titration curve. Bottom panels show $IC_{50}$ values of independent experiments (mean ± SD). Raw gMFI values are provided in the Source Data file. **h**, **i** Assessment of two IVIG preparations with reactivity to SARS-CoV-2 wildtype spike analogous to **d**–**g** (*n* = 3 experiments, performed in duplicates). **d**–**i** For statistical evaluation Shapiro-Wilk test was performed to assess normality of the data. $IC_{50}$ values of each variant were then compared to $IC_{50}$ values of other variants by repeated measure one-way ANOVA (**d,f-i**) or Mixed effect analysis (**e**) with Tukey's multiple comparison test. (*** *p* = 0.0001, **** *p* < 0.0001). Source data for this figure are provided in the Source Data file.

based on type II transmembrane proteins and demonstrate that they function with a signaling domain from the type I transmembrane protein CD3ζ. A potential limitation of our assay is that the presence of ligands for the chimeric receptors on the cellular biosensors themselves induces constitutive activation thereby impairing the sensitivity of the system for ligands presented in trans. For UL11, a CMV-derived surface molecule binding to CD45 we overcame this problem by using Jurkat reporter cells where CD45 expression was deleted using CRISPR/Cas9. We show that PD-1-ζ, PD-L1-ζ and 4-1BBL-ζ biosensors were triggered by anti-CD3/CD28 activated primary human PBMCs, reflecting the surface receptor/ligand expression pattern of these cells. Therefore, the biosensor technique could potentially be exploited for the identification of cells that express binding partners for ligands or receptors without known interaction partner, so called orphan receptors. We generated biosensors representing the ectodomains of four molecules that are considered orphan receptors or ligands and might interact with activated T cells (B7-H3, B7-H4, Butyrophilin 3A1 and CD5). In coculture experiments, none of these biosensors was activated in presence of resting or anti-CD3/CD28 activated PBMCs, whereas biosensors representing PD1 and 4-1BBL were strongly activated. These results indicate that PBMCs stimulated under the conditions used in our experiments do not express significant amounts of interaction partners for any of these orphan molecules.

The viral life cycle includes entry into a host cell followed by replication within the cell and finally shedding of new virions that go on to infect further cells. Therefore, the first step of a viral infection is the attachment of surface proteins presented on virions to host cell receptors. The expression pattern of host cell receptors in various tissues determines viral tropism. Many of such pairs of molecular interaction partners have been uncovered previously, including the binding of HIV gp120 to CD4, CXCR4 and CCR5 or the binding of measles virus hemagglutinin to CD150 and CD46[40,56,57]. More recently, in the beginning of the ongoing COVID-19 pandemic attachment of SARS-CoV-2 to ACE2 on respiratory tract epithelial cells was confirmed[44].

Based on these well-characterized host-virus interactions, we show that cellular biosensors expressing engineered virus entry receptors respond to the presence of the cognate virus attachment proteins with high sensitivity. A major mechanism of adaptive immunity to defend against viral infection is the B cell-mediated production of neutralizing antibodies, i.e., antibodies that bind the viral surface proteins in a way that the entry of the virus is blocked. Because of their functional role in immunity, neutralizing antibodies have been closely correlated with protection against viral infection[58,59]. The gold-standard to detect neutralizing antibodies are live-virus neutralization assays, which have been established for multiple virus species including measles, HIV and SARS-CoV-2[58,60,61]. These assays necessitate the use of live virus in specialized biosafety laboratories. Since the required facilities are not widely available, there is an urgent need for flexible and easy-to-handle assays to detect neutralizing antibodies. Currently used surrogate assays for neutralizing antibodies to viruses commonly rely on non-hazardous viruses pseudo-typed with the viral surface protein of interest or the detection of viral protein-host receptor interaction using cell-free immunological methods, e.g. ELISAs or electrochemiluminescence

immunoassays[47,61–63]. Results from these approaches closely correlate to neutralization assays, but frequently require laborious and costly production of recombinant proteins representing virus envelope proteins or pseudo-typed virus particles. When using recombinant proteins, validation of the proper folding and functionality may be necessary additionally, making high throughput analysis of different virus surface receptor variants difficult to achieve.

We show that the biosensors can be employed to study the interaction of the measles virus protein hemagglutinin with its cellular receptor CD46. We have also established a biosensor-based assay for CD4-HIV-1-gp160 interaction and validated this test with a panel of well-defined neutralizing antibodies to HIV-gp160 and CD4. HIV-1 is characterized by an exceptionally high mutational rate and this virus can rapidly acquire mutations in its envelope glycoproteins to escape neutralizing humoral immune responses and multiple rounds of neutralization escape are thought to occur in the infected host[64]. This mastery is regarded as the main reason for failures in attempts to develop effective HIV-1 vaccines[65]. Dissecting antibody neutralization and escape by HIV-1 as well as the evaluation of antibody quality of HIV-1 vaccines critically relies on the availability of effective and flexible neutralization tests that allow for high-throughput screening. We propose that biosensor-based assays fulfil these requirements since they afford the quick integration of novel surface gp120 and transmembrane gp41 variants.

Since the beginning of 2020 there is an ongoing pandemic with the novel coronavirus SARS-CoV-2. Antibody responses against the spike protein of this virus are of great interest since they are the major correlate for protection against re-infection and therefore necessary for a transformation of the pandemic to an endemic state[59]. We have developed a biosensor-based ACE-2-SARS-CoV-2-spike protein interaction assay for the detection of blocking antibodies. A high concordance of our biosensor-based method with a live-virus NT was observed. Our data indicate that this assay has the potential to be established as surrogate assay for the detection of SARS-CoV-2 neutralizing antibodies. Novel variants of this virus such as the Omicron variants impose an ongoing challenge because they are capable of escaping the immune system through mutations acquired within the spike protein which lead to reduced binding of antibodies and attenuated neutralization of the virus[66]. Biosensor-based surrogate neutralization assays are highly suitable to monitor protection against emerging variants since they do not require the time and laborious generation and validation of recombinant proteins representing the mutated virus attachment proteins. Since this assay is based on the stable expression of virus proteins and host receptors on human cell lines, proper folding and processing of the proteins is ensured. The assay relies on stable cell lines and therefore all components are fully replenishable. There is no requirement for additional reagents, which allows for inexpensive, high-throughput testing. We have adapted our assay for five SARS-CoV-2 variants (B.1.617.2, AY.4.2, B.1.1.529 BA.1, BA.2 and BA.5) and confirmed the differential capacity of therapeutic antibodies to block the interaction of these variants with ACE2. Therapeutic IVIG preparations are pooled from thousands of donors and could potentially provide insight into the overall neutralization capacity against emerging virus variants in the

donor population. We show that unlike monoclonal antibodies, IVIG preparations retained some neutralization capacity against all tested variants, hinting to a potential advantage of well characterized IVIG preparations over monoclonal antibodies in neutralizing emerging variant strains. The analysis of IVIG preparation and of sera of immunized and convalescent individuals indicates that the antibody immune escape of emerging variants is reflected on a populational level (pooled sera in IVIG preparations), while some individuals might still retain high neutralizing capacity.

In summary, we describe the development of cellular biosensors for receptor-ligand interaction assays. We show that our biosensors can be applied to study the interaction between different types of membrane-resident molecules as well as agents that block such interactions such as immune checkpoint inhibitors or neutralizing antibodies directed against virus attachment proteins. We further propose that the approach described here has applications that go beyond our current study. Mutations in virus receptors such as ACE2 have been shown to affect its interaction with virus attachment proteins[67,68]. Biosensors based on mutant ACE2 molecules could be used to explore the interaction with different SARS-CoV-2 Spike proteins as well as the capacity of antibodies and sera to block such interactions. The biosensor assay could aid the identification of cell populations that express interaction partners for orphan receptors or ligands thereby aiding their de-orphanization. Moreover, screening approaches using biosensors expressing mutational libraries of chimeric receptors could be used to identify molecules with improved binding profiles e.g. preferential interaction to certain ligands or to predict antibody or drug escape mutations in virus attachment proteins and membrane-resident tumor therapy targets, respectively.

## Methods

### Ethical statement
The study with human sera was approved by the ethics committee of the Medical University of Vienna under the registration number 2262/2020. Human PBMCs were obtained under approval by the ethics committee of the Medical University of Vienna under the registration number 1183/2016 as described previously[69]. Procedures with human material were performed in accordance with ethical standards of the ethics committee and the Helsinki Declaration of 1964 and its later amendments. Blood samples were collected from healthy volunteer donors after their informed consent.

### Study population
The cohort of serum donors was recruited at the FH Campus Vienna (n = 193) and the Institute of Immunology of the Medical University of Vienna (n = 18/ n-total =211). Donors at the FH Campus Vienna were analyzed for the presence of SARS-CoV2 specific antibodies using the electrochemiluminescence immunoassay (n = 193). Serum of 181 of the 193 FH Campus donors was available for further analysis with the biosensor assay. Age of serum donors was available for 193 of 211 donors (median 22a, range 18-61a). Age of PBMC donors was not reported. Furthermore, for some donors self-reported "antigen exposure" (contact, vaccination, self-reported testing) prior to sampling was assessed. Of all serum donors 63 had a reported contact to a SARS-CoV2 positive person prior to sampling. 126 had no contact, data of 22 donors was unavailable. 73 donors had a reported vaccination against SARS-CoV2 prior to sampling, 134 were unvaccinated, 3 did not report vaccination status. 32 donors reported having tested positive for SARS-CoV2 (testing methodology not specified) prior to sampling. Most donors were sampled once, 9 donors were sampled repeatedly (2-7 times). For Fig. 6g, SARS-CoV2 immunization status was obtained from one donor at 3 time points. For Fig. 6h, i serum samples were tested to compare the results from the biosensor assay to established assays and therefore any covariate would have influenced both assays simultaneously. For Fig. 4g, measles immunization status was obtained from one donor.

### Engineering of biosensor molecules
Biosensor molecules were cloned into a lentiviral expression vector based on the pHR-SIN-BX-IRES-Emerald (pHR Puro) vector[70]. The chimeric biosensor molecules consisted of a receptor extracellular domain fused to a CD28-transmembrane (TM) domain (UniProtKB P10747-1, aa153-179) and intracellular signaling domains of CD3ζ (UniProtKB P20963-3, aa52-163) or 4-1BB-CD3ζ (UniProtKB Q07011-1, aa214-255 and P20963-3, aa52-163). In some experiments, a truncated, signaling deficient PD-1 intracellular domain served as a non-signaling negative control domain (ctrl, UniProtKB Q15116-1 (aa 192-208)).

The cloning process required amplification of ectodomain-sequences from cDNA expression libraries[71,72] or existing expression vectors by PCR. XhoI and MluI cleavage sites were attached to the 5' and 3' ends through appropriate primer design (Cloning primers are listed in Supplementary Table 1). The inserts were then ligated into a XhoI/MluI digested pHR Puro vector encoding a downstream CD28-TM domain, the respective signaling domain and a BamHI site. A P2A ribosomal skipping site followed by a gene encoding for puromycin-N-acetyl-transferase was located 3' of the BamHI site and allowed for selection of successfully transduced cells. The UL11-, and ACE2-inserts and BTN3A1-ζ were designed in silico and synthesized (TWIST Bioscience, CA, USA).

To clone type II transmembrane biosensor molecules, overlap extension PCR was used to create chimeric molecules as indicated in Supplementary Table 2 with XhoI/BamHI cleavage sites at the 5' and 3' ends. The inserts were then ligated into the XhoI/BamHI digested pHR Puro vector.

Sequences for measles hemagglutinin and HIV-1 gp160 were derived from the Addgene (Cambridge, MA) plasmids pCG-HcΔ18[73] and pCEP-BaLgp160[74], respectively. For the SARS-CoV-2-spike protein variants consensus mutations were inserted manually. The constructs were synthesized (TWIST Bioscience) and either cloned into pHR Puro or ordered directly cloned into the lentiviral expression vector pTWIST Lenti SFFV Puro WPRE (TWIST Bioscience).

Sequence accuracy was confirmed for every construct by Sanger sequencing (Eurofins Genomics, Luxembourg). Supplementary Table 2 lists all unmodified, modified and chimeric constructs by name and amino acid sequence used.

### Cell lines, media, reagents, and flow cytometry
Jurkat E6.1 (JE6.1), murine BW5147 (short designation within this work: BW) and K562 cell lines were cultured in RPMI 1640 supplemented with 100 µg/ml streptomycin, 100 U/ml penicillin, 2.5 µg/mL amphotericin B and 10% heat-inactivated fetal calf serum (FCS) (Thermo Fisher Scientific, MA, USA) at 37 °C and 5% CO₂. HEK293 cells were kept in IMDM medium supplemented as the RPMI-1640 medium with addition of 2mM L-glutamine. For this study, the previously described reporter cell lines JE6.1 NFκB-eGFP and JE6.1 TPR were used[19,20]. These T cell lines retain a functional TCR signaling pathway and are equipped with reporter genes that upon binding of the transcription factor NFκB or NFκB, NFAT and AP-1 induce expression of the fluorescent proteins eGFP or eCFP, eGFP and mCherry respectively. Another reporter cell line used in this study were JE6.1 Nur77-mKO2-reporter cells that have been described previously[75]. These cells harbor an in-frame knock-in of a T2A-mKusabira-Orange2 (mKO2) module at the Nur77 gene generated by homology directed repair of CRISPR/Cas9 induced DNA double strand breaks. Therefore, upon transcription of Nur77 the cells express the fluorescent protein mKO2. Additionally, JE6.1 Nur77-mKO2 reporter cells had a CRISPR/Cas9 mediated knockout of the *TRAC* and *TRBC* genes (*TRAC/TRBC*ᴷᴼ-JE6.1-Nur77-mKO2) and expressed the fluorescent protein mAmetrine for identification in co-culture assays. Several chimeric biosensor molecules were tested in more than one reporter cell line. The cell lines that have been used for the experiments are indicated in the figure legends. The Nur77-reporter cells are available from M.L. with a completed material transfer agreement. All other cell lines

generated in this study are available from the corresponding authors upon completion of a material transfer agreement.

Chimeric receptor molecules were delivered into reporter cells by lentiviral transduction. BW and K562-RFP cell lines were used as stimulator cells by expressing the respective biosensor molecule ligands through lentiviral or retroviral transduction. As positive control K562 expressing a membrane bound anti-CD3 scFv were used in some interaction assays (K562 anti-CD3). For some experiments BW expressing mCD32B pre-incubated with an antibody binding this Fc-receptor were used as stimulator cells. In some experiments primary human PBMCs were activated for 1-6 days using anti-CD3 (clone OKT3, Johnson & Johnson, NJ, USA) and anti-CD28 (clone CD28.2, Biolegend, CA, USA) coated Dynabeads (Thermo Fisher Scientific) and IL-2 (50 IU/ml, Peprotech, NJ, USA).

To select for successfully transduced cells the cell culture medium was supplemented with $1 \mu g/ml$ puromycin. To detect expression of the respective molecules we used the fluorophore-conjugated antibodies listed in Supplementary Table 3. Cells were incubated with the respective antibodies for 20 minutes at $4\,°C$ in concentrations according to the manufacturers' recommendation. Flow-cytometric measurements were performed on a FACS Calibur with CellQuest Pro software Version 6.0, a LSRFortessa with FACSDiva software Version 9.0 (both BD Bioscience, NJ, USA) or a CytoflexS with CytExpert software Version 2.4 (Beckmann Coulter, CA, USA). Further analysis of the flow cytometry data was then performed using FlowJo 10.7.1 (BD Bioscience, NJ, USA). Flow sorting was performed on a Sony SH800 (Sony Biotechnology, CA, USA). FACS data is shown as gMFI (geometric mean fluorescence intensity). For some experiments, reporter gene induction in response to stimulation was normalized to unstimulated reporter cells as indicated and expressed as fold induction (FI). For some cell lines we additionally performed single cell cloning to select high expression clones. For mycoplasma detection $50 \mu l$ of cell free culture supernatants were applied to $5 \times 10^4$ THP-1 NFκB-eGFP reporter cells in a final dilution of 1:2. After 24 h the samples were analysed by flow cytometry and cell cultures whose supernatants induced eGFP gMFI values that were 50% higher compared to untreated reporter cells were scored as mycoplasma positive[76].

## CRISPR/Cas9

To avoid interference of molecules naturally expressed on the cell lines we used CRISPR/Cas9 mediated knockout of the respective molecules. We performed knockout of *CD46* and *CD4* in JE6.1 TPR and knockout of *PTPRC* (encoding CD45) in JE6.1 TPR and K562 RFP. For *PTPRC* and *CD46* knockout predesigned single guide RNAs (*PTPRC*: 5′-ACAACCACTCT-GAGCCCTTC-3′, *CD46*: 5′-GCAAATGGGACTTACGAGTT-3′) together with tracrRNA and Cas9 was purchased from Integrated DNA Technologies (IDT, IO, USA). Ribonucleoprotein complex was prepared according to the manufacturers protocol. $5 \times 10^5$ Jurkat JE6.1 TPR reporter cells were electroporated with the Neon Transfection System (Thermo Fisher Scientific) using settings recommended by the manufacturer. Successfully knocked out cell lines were sorted by negative fluorescent-antibody staining of CD45 and CD46 respectively. To create *CD4* knockout reporter cells, JE6.1 TPR reporter cells were lentivirally transduced with the lentiCRISPR v2 vector purchased from Addgene (Cat. 52961) coding for Cas9 and for a guide RNA targeting *CD4* (5′-GTCTGTAAAACGGGTTACCC-3′). To select the single-guide sequence the CRISPR design website ATUM (http://www.atum.bio) was used. The guide RNA was inserted according to the protocol described by Sanjana et al. [77]. Briefly, phosphorylated oligonucleotides encoding the sgRNA guide sequence were annealed by heating to $95\,°C$ for 5 minutes and cooling to $25\,°C$ at a rate of $1.5\,°C/minute$. The sgRNA insert was cloned into the lentiCRISPR v2 vector via BsmBI sites and correct insertion was confirmed by DNA sequencing. The resulting vector and virus packaging plasmids were transfected into HEK293T and cell culture supernatant containing lentiviral particles was used to transduce JE6.1 TPR

reporter cells[78]. Single cell cloning of the transduced cells was performed and a CD4 negative clone identified through fluorescent-antibody staining. All knockouts were additionally confirmed by Sanger sequencing (Eurofins Genomics) of target site PCR products amplified from genomic DNA prepared using the Gentra Puregene® Cell Kit (Qiagen, Hilden, GER). Knockout efficiency was then confirmed using the TIDE online tool (http://shinyapps.datacurators.nl/tide/)[79].

## Transient expression

For transient expression of ligand molecules on HEK293 cells calcium phosphate precipitation with the respective expression vectors was used. HEK293 cells were seeded at $1.5 \times 10^5$ cells per $500 \mu l$ in a 24-well plate. After 24 h supernatant was replaced with 1 ml of fresh medium. $2 \mu g$ of expression vector was co-incubated with 2,5 M $CaCl_2$ in $50 \mu l$ $H_2O$. Next, $50 \mu l$ of 2x HBS-Buffer (pH 7.05, 140 mM NaCl, 1.5 mM Na2HPO4, 50 mM HEPES) was added while vortexing, followed by incubation for 1 min. Finally, the mixture was added to HEK293 cells. For expression of SARS-CoV2-spike the pcDNA3.1-SARS2-Spike[80] vector was acquired through Addgene (Cat. 145032). A pcDNA B7H3 vector and treatment without an expression vector served as negative controls.

## Receptor-ligand interaction assays using a Jurkat-reporter system

If not otherwise specified $5 \times 10^4$ reporter and $2 \times 10^4$ stimulator cells were mixed in a 96-well flat bottom plate and cultured for 24 h at $37\,°C$ and 5% $CO_2$. The next day cells were harvested, transferred to microtiter tubes, washed with PBS supplemented with 0.5 % FCS and 0.1 % sodium azide and measured by flow cytometry. K562-based stimulator cells were identified by RFP-expression. BW-based stimulator cells were excluded from analyses using an anti-mouse-CD45.2-APC antibody. Fluorescence intensity of reporter cells, defined as viable, RFP/APC negative cells, was then measured by flow cytometry. Reporter cell activation was expressed as an increase in geometric mean fluorescent intensity of the respective reporter gene product over unstimulated reporter cells. Blocking agents such as sera, intravenous immunoglobulin preparations, antibodies, fusion proteins and small molecule inhibitors (listed in Supplementary Table 4) were preincubated with stimulator cells for 30 min at room temperature before initiation of the co-culture. SIM.2, CH106, N6-PGDM1400x10E8, VRC-CH31, VRC-01, CD4-Ig, NIH45-46 G54W, PGT128 were acquired through the NIH HIV Reagent Program (NIH HRP), Division of AIDS, NIAID, NIH.

## Co-culture assay with activated PBMCs

Primary human PBMCs were isolated by standard density gradient centrifugation using the Lymphoprep reagent (Technoclone, Austria).

For the assay to identify ligands to orphan receptors, resting PBMCs (d0) and PBMCs activated with anti-CD3/anti-CD28 (each used at $1 \mu g/ml$ for coating) coated Dynabeads (used at a ratio of 1:1) and 50 U/ml IL-2 were used. PBMCs were harvested at the indicated time-points and beads were removed by magnetic separation prior to co-culture with reporter cells. PBMCs (used at a final concentration of $5 \times 10^5/ml$) and reporter cells were co-cultured at a ratio of 1:1. Following 24 h of co-culture, activation of the reporter gene Nur77-mKO2 was assessed by flow cytometry.

## Microscopy

To acquire fluorescent-microscopy images of the JE6.1 ACE2-ζ reporter cell line a receptor/ligand interaction assay was set up in a 96-well plate as described earlier. Representative images of each well were acquired using an Evos M500-fluorescence-microscope (Thermo Fisher Scientific). Subsequently technical replicates from the same assay were analyzed by flow cytometry to calculate geometric mean fluorescent intensity of each condition.

## Electrochemiluminescence immunoassay

To quantitatively determine serum antibody levels against the RBD of the SARS-CoV-2 spike protein the cobas® based Elecsys Anti-SARS-CoV-2 S assay (Roche Diagnostics, Rotkreuz, CH) was used. The assay was performed according to the manufacturer's recommendation. Briefly the assay relies on a double sandwich immunoassay principle. A biotinylated, recombinant RBD-antigen and a Ruthenium-complexed RBD-antigen are co-incubated with analyte serum. Both antigens are crosslinked in the presence of RBD-specific antibodies. Streptavidin-coated microparticles are added. They bind the biotinylated part of the immunocomplexes which are then fixed to an electrode by the microparticles. Through the application of voltage, a chemiluminescence reaction is induced and light emission can be measured via a photomultiplier. Results are then fitted to a predefined standard curve. All electrochemiluminescence immunoassays were run on a cobas® e411 analyzer (Roche, Mannheim, GER).

## SARS-CoV-2 neutralization assay

The SARS-CoV-2 live-virus NTs were performed as described previously[81]. Briefly, heat-inactivated serum samples were serially two-fold diluted and incubated in duplicates with 50-100 TCID50 SARS-CoV-2 (B.1.1 with the D614G mutation: EPI_ISL_438123) for 1 h at 37 °C. The mixtures were then added to Vero E6 cells (ECACC #85020206). After incubation for three days at 37 °C, cytopathic effect was microscopically evaluated. NT titers were determined as the highest reciprocal serum dilution at which no cytopathic effect was present. NT titers ≥10 were considered positive.

## Surface molecule quantification

To quantify surface expression of CD80 and CD86 on K562 stimulator cells we used the QUANTUM-R-PE MESF kit (Bangs Laboratories Inc.). K562 CD80 and K562 CD86 cells were stained with CD80-PE and CD86-PE antibodies, respectively, and the assay was performed according to the manufacturers' instructions.

## Statistics & reproducibility

Statistical analysis was performed using Prism 9 (GraphPad, MA, USA). For each population normality of distribution was assessed using the Shapiro-Wilk test. To compare means of multiple, populations the following statistical tests were applied depending on experimental setup and normality of the dataset. For unpaired data ordinary one-way ANOVA with Dunnett's multiple comparison test or Kruskal-Wallis test with Dunn's multiple comparison test was used for normally and not normally distributed data, respectively. To compare paired data repeated measure one-way ANOVA with Tukey's multiple comparisons test or Friedman's test with Dunn's multiple comparisons test for normally and not normally distributed data, respectively. In one experiment a mixed effects model (REML) with Tukey's multiple comparisons test was used for a dataset with missing values because one replicate could not be analyzed. For comparisons of multiple groups regarding two variables two-way ANOVA with Šídák's multiple comparisons test was used. We performed nonlinear regression using the Sigmoidal, 4PL, X is log(concentration)-equation (constraints: bottom to 0 and top to 1) to fit titration curves of small molecule inhibitors, IVIG preparations, neutralizing antibodies and sera. For correlation of two non-normally distributed populations we used Spearman's rank correlation. No statistical method was used to predetermine sample size. The experiments were not randomized. The investigators were not blinded to allocation during experiments and outcome assessment.

## Creation of Schemes

Schemes (Figs. 1a, f; 2a, c; 3a, g, h; 4a, d; 5a, c; 6a, b; Supplementary figures 2e; 3a, c; 6a, c, g.) were created using BioRender (Toronto, CA).

## Reporting summary

Further information on research design is available in the Nature Portfolio Reporting Summary linked to this article.

## Data availability

A Source Data file containing all raw geometric mean fluorescence intensity (gMFI) values, normalization steps and statistical tests is provided as supplementary file with this paper. Supplementary Table 2 lists the UniProtKB entries of all proteins expressed in cell lines for this study. All amino acid modifications are also listed in Supplementary Table 2. Source data are provided with this paper.

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

## Acknowledgements

This study was supported by the Austrian Science Fund (FWF-P32411) to PS, a grant by the City of Vienna Fund for Innovative Interdisciplinary Cancer Research to MF (22025/2021) and a grant by the Medical Scientific Fund of the Mayor of the City of Vienna to KGP (21130/2021). BS was supported by the Federal Ministry for Digital and Economic Affairs of Austria and the National Foundation for Research, Technology and Development of Austria to the Christian Doppler Research Association (Christian Doppler Laboratory for Next Generation CAR T Cells). We wish to thank Claus Wenhardt and Jutta Hutecek for technical assistance. The sequence of a SARS-CoV-2-spike protein stabilized for cell surface expression was provided by Mirjam Klausberger, University of Natural Resources and Life Sciences Vienna Austria. We are grateful to the NIH HIV Reagent program for providing antibodies to CD4; HIV gp120 and gp160.

## Author contributions

M.F. performed the majority of experiments, designed the study, analyzed data and wrote the manuscript. J.L. performed experiments with PD-L1-ζ reporter cells and with PBMC samples, analyzed data and wrote the manuscript. C.B. performed experiments with PD-1-ζ reporter cells. K.S. contributed the SARS-CoV-2 neutralization assay data. M.C.G. and J.M.H. performed and supervised the electrochemiluminescence immunoassays and provided serum samples. B.S. and M.L generated Nur77-mKO2 reporter cells. K.G.P. performed experiments and provided essential reagents. S.G. carried out some experiments and provided technical support. P.S. designed and supervised the study and wrote the manuscript. All authors critically revised the manuscript and approved of the final version.

## Competing interests

The authors declare no competing interests.
