## [Peer Review File · Nature Communications]

REVIEWER COMMENTS

Reviewer #1 (Remarks to the Author):

The manuscript by Funk et al. describes a cell-based system to detect and quantify cell surface receptor ligand interactions. They describe this as a cellular biosensor which is based on the fact that extracellular binding events and intracellular signalling can be coupled by making a chimeric receptors containing different extracellular regions and signalling-inducing intracellular regions whose activity can be quantified using activation of a GFP reporter gene and flow cytometry readout. The authors show that the system is specific and sensitive in a range of different contexts using the PD1/PD1L, 4-1BBL/4-1BB endogenous receptor-ligand systems and also for host- viral interactions (UL11/CD45, CD46/measles Hg, CD4/HIV-1gp160, SARS S, ACE2). They demonstrate specificity and the usefulness of the system using a range of blocking antibodies targeted to the signal sending cell. The authors also claim that it can be used to identify ligands for orphan receptors and show a proof of concept experiment; however, they do not identify a new receptor-ligand pair in this manuscript.

Overall this is a well-executed and well-written manuscript. It describes a well-characterised version of an established system and so the paper does suffer from a lack of novelty. Using chimeric receptors to discover binding events at the cell surface is not a new idea and so the paper describes a technical refinement rather than anything particularly new. The authors suggest that this assay could be used to identify binding partners for “orphan” receptors but they should take that next step and demonstrate this. As the authors acknowledge, there is a limitation that ligands present of the cellular biosensors themselves could induce constitutive activation (either in cis or trans) which could prevent its use for this purpose.

Major points:

The authors should show the raw FACS profiles from the activated signalling rather than the processed gMFI. This applies to Fig1C, 1H, 2D, 2J etc...

The authors should show the data to support their statement “The signal could be further enhanced by co-expressing CD86 on gp160+ cells”

Minor errors:

Typo: MPOC-21 should be MOPC-21 (Table2)

Typo “serum antibodies levels” should be “serum antibody levels”

Please check amino acids numbers: “and introducing the K1043P and V1043] P mutations”

Text is too small to read on some figures e.g. concentrations on x-axis e.g. in Fig.1

Reviewer #2 (Remarks to the Author):

This is in essence a very straightforward paper. The authors have established a simple, sensitive assay for detecting the interactions of proteins that can be expressed at the surfaces of T-cells, or at least on cells capable of responding to the phosphorylation of the cytosolic region of the T-cell receptor (TCR). The assay comprises expressing, on the “biosensor” cell, a chimeric “receptor” comprising the protein of interest (POI), coupled via a transmembrane sequence to the cytosolic region of the TCR. Interactions of the extracellularly-expressed POI, with cell expressed “ligands” of the POI lead to signaling in the biosensor cell. In effect, the method comprises a straightforward extension of the chimeric antigen receptor principle to other receptor ligand pairs.

The authors exemplify the utility of the method in a variety, if not exhaustive array of settings, along the way producing some nice vignettes. These include (1) showing that the PD-1/PD-L1 interaction produces signaling in both orientations, suggesting that there are no receptor-specific structural requirements of the triggering mechanism whatever that is, (2) further demonstrating that PD-L1 and CD80 interact only very weakly in trans, and (3) showing that the biosensor can detect even extremely weak interactions, for example those between CD4 and MHC class II molecules. The authors don't otherwise systematically explore the requirements of the assay, e.g., the dependence of the assay on expression level or affinity of interaction, or what other structural factors could affect the strength of the readouts or general utility of the assay. The surprisingly-weak signal generated by CD4 interacting with gp160 of HIV-1 could perhaps be suggesting that the readout might rely on the dimensions of the interacting protein pair, implying in turn that it will work best (or only) for small proteins. Some consideration of current ideas about receptor triggering that could be relevant to the utility of the assay could perhaps have been considered in the discussion.

The experiments are well thought out and carefully done as is typical of this laboratory, and the manuscript well written. I have no concerns whatsoever about the reliability of the analysis or the conclusions. A slight caveat is whether it needed to be quite so long; a lot of the data could perhaps have gone to the SI. To this I would only add the slight concern about whether it's very novel, given all the work that's been done on, especially, chimeric antigen receptors. Some would consider this to be an

obvious extension of that technology. On the other hand, I'm impressed by the sensitivity of the assay and how faithfully it "reads out" in trans versus in cis interactions. The case for this being a useful assay is therefore well made, I think.

Reviewer #3 (Remarks to the Author):

Maximilian Funk and colleagues established cellular biosensors as alternative tool to study ligand-receptor interactions. Using this system, the authors evaluated immune checkpoint inhibitors, and neutralizing antibodies to HIV and SARS-CoV-2 including emerging variants. While the work is some interest with extensive interactions of membrane-resident proteins and the characterization of their inhibitors, there are several important points to be addressed.

Major issues:

1. Figure 1, the authors should also test the matched isotype control antibody in Fig 1D and 1G. The gMFI is above 1000 in Fig 1C when treated with BW PD-L1, but the same stimulation is only about 30 in Fig 1D. Please explain this discrepancy. The value of unstimulated control is also inconsistent in different panels. Could the assay characterize small molecule inhibitors of the PD-1/PD-L1 protein-protein interaction?
2. Figure 2, No signal attenuation is observed in Fig 2F. Utomilumab even promotes the signal intensity when treated with 0.1ug/ml. Please explain this result since Utomilumab is a humanized monoclonal antibody that activates 4-1BB while blocking binding to endogenous 4-1BBL, while Urelumab does not block the interaction of 4-1BB with its ligand (Blood,2018,131 (1): 49–57; Nat Commun. 2018,9(1):4679).
3. Figure7, when did the sera collect from donor 44C-46C (Fig. 7I-K)? Did they vaccinate with wild-type Spike? Why did individual serum samples show less neutralization capacity against wild-type SARS-CoV-2 than that of Omicron strain. This result is inconsistent with previous studies (N Engl J Med. 2022;387(1):86-88, PMID: 35731894; N Engl J Med. 2022,386(7):698-700, PMID: 35021005).
4. The authors claim that the biosensor-based interaction assay is a highly accurate surrogate test for the detection of SARS-CoV-2 neutralizing antibodies. However, I am concerned about the data comparability (J Clin Virol. 2022,156: 105292. PMID: 36108404). The authors display amount of data on neutralizing antibody titers that are not "normalized" and thus difficult to interpret without a reference

cohort. In addition, the expression levels of the respective molecules (ACE2, Spike) vary among different cell clones. How to ensure the antibody titer comparison between different laboratories and over time?

5. Infection with SARS-CoV-2 is initiated by virus Spike binding to the ACE2, followed by fusion of the virus and cell membranes (Nature,2020,588,327–330; Sig Transduct Target Ther 2020,5,92, PMID: 32532959). Several peptides and antibodies recognize the conserved epitope in the S2 subunit of the Spike can inhibit SARS-CoV-2 infection by blocking the Spik-mediated membrane fusion (Microbiol Spectr. 2022,10(2): e0181421. PMID: 35293796; Science,2021,371(6536):1379-1382.). Can small molecules, peptides or anti-S2 antibodies block such interactions and reduce gMFI in the biosensor cell assay?

6. As the authors point out, all their work is performed with a version of “wild-type” Spike protein without furin-cleavage site (RRRdel) and bearing the K1043P and V1043P mutations (Page 12 line 434). This could affect their data. Stimulator cells can be readily generated with full-length Spike protein, an analysis that should be performed.

7. How did author define the limit of detection of biosensor interaction assay? What is the quantitative range of this assay for SARS-CoV-2 neutralizing antibody detection?

Minor points:

1. For gene expression, please use the appropriate nomenclature (italic, etc.)
2. Page 4, Table 2, why did the authors introduce K986P and V986P mutations in SARS-CoV-2-spike?
3. Please provide a better resolution image. The font on Figures is too small, and the label needs to be clear.
4. Figure legends should be more informative for better understanding. Figure 1, please define BW and gMFI in the figure legend.
5. Figure 1H, please provide the expression levels of CD80 and CD86 in biosensor cells. Is there a significant difference between CD80 and CD86 groups (Fig. 1H)?

6. Page 10 line 323: "As few as 2500 stimulator cells (reporter-to-stimulator ratio, 20:1)" In Fig 2E legend, 5×10^5 reporter cells were used, the ratio should be 200:1?

7. The legends for Fig 2F and 2G are reversed.

8. Please define mKO2 in line 414.

9. Figure 7, since the D614G was not observed in the Spike of Wuhan strain, did the author use different wild-type Spike proteins in 7A and 7B?

10. For SARS-CoV-2 neutralization assay, how much virus dose was used? How is a live virus titrated? Please also define the neutralization titer in detail and provide the NT50 values.

11. Could the author provide a workflow of the receptor-ligand interaction assay, including timing?

12. The receptor-ligand interaction assays in the method section requires more details. It will be useful to have more details on critical steps.

REVIEWER COMMENTS

Reviewer #1 (Remarks to the Author):

The manuscript by Funk et al. describes a cell-based system to detect and quantify cell surface receptor ligand interactions. They describe this as a cellular biosensor which is based on the fact that extracellular binding events and intracellular signalling can be coupled by making a chimeric receptors containing different extracellular regions and signalling-inducing intracellular regions whose activity can be quantified using activation of a GFP reporter gene and flow cytometry readout. The authors show that the system is specific and sensitive in a range of different contexts using the PD1/PD1L, 4-1BBL/4-1BB endogenous receptor-ligand systems and also for host- viral interactions (UL11/CD45, CD46/measles Hg, CD4/HIV-1gp160, SARS S, ACE2). They demonstrate specificity and the usefulness of the system using a range of blocking antibodies targeted to the signal sending cell. The authors also claim that it can be used to identify ligands for orphan receptors and show a proof of concept experiment; however, they do not identify a new receptor-ligand pair in this manuscript.

Overall this is a well-executed and well-written manuscript. It describes a well-characterised version of an established system and so the paper does suffer from a lack of novelty. Using chimeric receptors to discover binding events at the cell surface is not a new idea and so the paper describes a technical refinement rather than anything particularly new. The authors suggest that this assay could be used to identify binding partners for “orphan” receptors but they should take that next step and demonstrate this. As the authors acknowledge, there is a limitation that ligands present of the cellular biosensors themselves could induce constitutive activation (either in cis or trans) which could prevent its use for this purpose.

Response to reviewer 1

We wish to thank this reviewer for her/his insightful comments. As pointed out in the introduction of the manuscript our work is based on pioneering work by others that have developed chimeric receptors and demonstrated their usefulness to redirect effector cells to target cells of interest such as virus infected cells or tumor cells. However, we feel that our study does introduce novel aspects of the use of chimeric receptors. We introduce the concept of integrating chimeric receptors into sensitive reporter cells and use the resulting “biosensors” to study receptor-ligand interactions (whereas earlier studies have mainly used chimeric receptors to re-target effector populations). Furthermore, the use of chimeric receptors based the ectodomain of virus receptors to study its interaction with virus entry molecules and to effectively assess the neutralization capacity of therapeutic antibodies or antibodies contained in sera of patients or vaccinees has not been reported previously. The use of our biosensor system to detect binding partners for orphan ligands and receptors is also a novel concept. However as rightfully criticized by this reviewer this matter our study initially did not go beyond proof of concept experiments. For the revision of our study we have undergone extensive efforts generate biosensors based on four different orphan receptors and ligands (CD5; B7-H3; B7-H4 and Butyrophilin 3A1) and probed them with resting and in vitro activated PBMC samples. The integrity and functionality of chimeric receptors harboring the ectodomains of these molecules was confirmed by using specific antibodies for staining and in conjunction with Fc-gamma IIB receptor bearing cells. Biosensors harboring the ectodomains of these orphan molecules were strongly activated by antibodies demonstrating activation upon ligation (Figure S4d-g). However they did not activate at all in co-cultures with resting and CD3/CD28 activated PBMC samples from different donors. By contrast biosensors based on PD1 or 41BBL showed very strong activation with PBMC samples for all donors (Figure 3g-j of the revised manuscript). We also wanted to include biosensor cells harboring a chimeric receptor that should not interact with molecules expressed on human PBMCs. We chose a chimeric receptor based on murine GITR (mGITR ζ) since binding assays with fusion proteins of human and murine members of the TNF-SF and cells expressing high levels of

human and murine TNFR-SF did not detect interaction between murine GITR with the human orthologue of its ligand – GITRL whereas there was reactivity between most murine members of TNFR-SF with the human orthologues of their respective ligands as shown by a study of the group of Pascale Schneider (<https://doi.org/10.1074/jbc.M601553200>). Interestingly, the mGITR- ζ biosensor intended as a negative control was clearly activated by PBMC samples from all three donors (Figure 3j) and subsequent experiments confirmed a specific interaction of mGITR ζ with human GITRL (Figure S5). Thus although our experiments have not identified cell populations that harbor binding partners for surface molecules that have been implicated as receptor or ligands for a of yet unknown molecules (which is almost impossible within the time available for the revision of a manuscript) they corroborate the utility of our biosensor approach for identifying cell populations expressing ligands for orphan molecules by showing its high sensitivity which allow the detection of interactions that have gone undetected in previous studies (mGITR – hGITR-ligand) and also the lack of background activation of the biosensors. Although our results do not rule out the existence of binding partners for the ectodomains of CD5; B7-H3; B7-H4 and Butyrophilin 3A1 they indicate that PBMCs samples activated under the conditions used in our study are unlikely to express significant amounts of such molecules and strategies to identify such molecules should focus on other cell populations and differentially activated PBMC samples. We feel that these experiments provide important information and clearly strengthened our point that the biosensor approach can be used to investigate orphan molecules and thus considerably improved our manuscript.

Major points:

The authors should show the raw FACS profiles from the activated signalling rather than the processed gMFI. This applies to Fig1C, 1H, 2D, 2J etc...

Response to reviewer 1:

We have added FACS-profiles throughout the manuscript to show the activation of the reporter cells expressing biosensors in presence of ligand expressing cells: Figure S1b and c (PD-1-zeta); Figure 2d (4-1BBL-zeta); Figure S2v (PD-L1-zeta); Figure S3d (UL11-zeta); Figure S5d (mouse GITR-zeta); Figure 4e; g (CD46-zeta); Figure 5e and Figure S6d; f (CD4-zeta), Figure 6f, Figure 8b and Figure S7c (ACE2-zeta).

The authors should show the data to support their statement “The signal could be further enhanced by co-expressing CD86 on gp160+ cells”

Response to reviewer 1: ‘

We show the activation of CD4-zeta reporter cells in response to K562-gp160 and K562-gp160-CD86 in figure S6f of the revised manuscript.

Minor errors:

Typo: MPOC-21 should be MOPC-21 (Table2)

Typo “serum antibodies levels” should be “serum antibody levels”

Please check amino acids numbers: “and introducing the K1043P and V1043] P mutations”

Text is too small to read on some figures e.g. concentrations on x-axis e.g. in Fig.1

Response to reviewer 1:

These errors have been corrected and the aa numbers has been changed to “ K1043P and V1044P” – we wish to thank this reviewer for pointing them out to us. We have modified all figures to improve resolution and legibility.

Reviewer #2 (Remarks to the Author):

This is in essence a very straightforward paper. The authors have established a simple, sensitive assay for detecting the interactions of proteins that can be expressed at the surfaces of T-cells, or at least on cells capable of responding to the phosphorylation of the cytosolic region of the T-cell receptor (TCR). The assay comprises expressing, on the "biosensor" cell, a chimeric "receptor" comprising the protein of interest (POI), coupled via a transmembrane sequence to the cytosolic region of the TCR. Interactions of the extracellularly-expressed POI, with cell expressed "ligands" of the POI lead to signaling in the biosensor cell. In effect, the method comprises a straightforward extension of the chimeric antigen receptor principle to other receptor ligand pairs.

The authors exemplify the utility of the method in a variety, if not exhaustive array of settings, along the way producing some nice vignettes. These include (1) showing that the PD-1/PD-L1 interaction produces signaling in both orientations, suggesting that there are no receptor-specific structural requirements of the triggering mechanism whatever that is, (2) further demonstrating that PD-L1 and CD80 interact only very weakly in trans, and (3) showing that the biosensor can detect even extremely weak interactions, for example those between CD4 and MHC class II molecules. The authors don't otherwise systematically explore the requirements of the assay, e.g., the dependence of the assay on expression level or affinity of interaction, or what other structural factors could affect the strength of the readouts or general utility of the assay. The surprisingly-weak signal generated by CD4 interacting with gp160 of HIV-1 could perhaps be suggesting that the readout might rely on the dimensions of the interacting protein pair, implying in turn that it will work best (or only) for small proteins. Some consideration of current ideas about receptor triggering that could be relevant to the utility of the assay could perhaps have been considered in the discussion.

The experiments are well thought out and carefully done as is typical of this laboratory, and the manuscript well written. I have no concerns whatsoever about the reliability of the analysis or the conclusions. A slight caveat is whether it needed to be quite so long; a lot of the data could perhaps have gone to the SI. To this I would only add the slight concern about whether it's very novel, given all the work that's been done on, especially, chimeric antigen receptors. Some would consider this to be an obvious extension of that technology. On the other hand, I'm impressed by the sensitivity of the assay and how faithfully it "reads out" in trans versus in cis interactions. The case for this being a useful assay is therefore well made, I think.

Response to reviewer 2:

We wish to thank the reviewer for this thoughtful and encouraging comments. We agree with this reviewer that it would be definitely interesting to assess the sensitivity of our assay in more detail and this is one of our future plans for this assay. As indicated by this reviewer such experiments are very challenging since they might depend on the expressing levels of the interaction partners the affinity of the interaction and other factors such as molecule size and steric constraints as implied by this reviewer. To our knowledge the sensitivity of other assays for protein – protein interaction such as ELISA or SPR is not well defined despite the fact that these assays have been in use for many decades.

Also it is true that it appears to work especially well with rather small molecules, we do not think that the assay is limited to the interrogation of the interaction of small proteins since we demonstrate that it can be used to detect e.g. the interaction of SARS-CoV-2 spike and ACE2 which have both rather large ectodomain. We suspect that rather weak signal mediated by HIVgp160 is due to the low expression of gp160 and measures to improve its surface expression such as deletion of the recycling motifs in the cytoplasmic tails of gp160 might improve the signal. Further work is required to address this.

Regarding the novelty of our approach, I think it is frequently a matter of debate whether a methodology is a logical next step or a real innovation. We are aware that our study is based on

important previous studies of others and we have tried to give credit to all the researchers whose work was the basis of our study. Nevertheless, we feel that several aspects of our work are novel and innovative. We introduce the concept of integrating chimeric receptors into sensitive reporter cells and use the resulting “biosensors” to study receptor-ligand interactions which to our knowledge has not been done before since earlier studies have mainly used chimeric receptors to re-target effector populations. Furthermore, the use of chimeric receptors based on the ectodomain of virus receptors to study its interaction with virus entry molecules and to effectively assess the neutralization capacity of therapeutic antibodies or antibodies contained in sera of patients or vaccinees has not been reported previously. We think that the use of our biosensor system to detect binding partners for orphan ligands and receptors is also a novel concept and for the revision of our manuscript we have extended this part by generating and validating biosensors based on orphan molecules and probing them with resting and activated PBMCs obtained from different donors.

As suggested by this reviewer we have tried to shorten our manuscript by moving some of the data that were in the main manuscript of the initial submission (Fig. 2c-j; Fig. 5f-g and Fig. 7g) to the supplement. We think that this has considerably improved the readability of our manuscript.

Reviewer #3 (Remarks to the Author):

Maximilian Funk and colleagues established cellular biosensors as alternative tool to study ligand-receptor interactions. Using this system, the authors evaluated immune checkpoint inhibitors, and neutralizing antibodies to HIV and SARS-CoV-2 including emerging variants. While the work is some interest with extensive interactions of membrane-resident proteins and the characterization of their inhibitors, there are several important points to be addressed.

Major issues:

1. Figure 1, the authors should also test the matched isotype control antibody in Fig 1D and 1G. The gMFI is above 1000 in Fig 1C when treated with BW PD-L1, but the same stimulation is only about 30 in Fig 1D. Please explain this discrepancy. The value of unstimulated control is also inconsistent in different panels. Could the assay characterize small molecule inhibitors of the PD-1/PD-L1 protein-protein interaction?

Response to reviewer 3

We want to thank the reviewer to point out this potential unclarity. Within this study three different flow cytometers were used (FACSCalibur, LSRFortessa, CytoFLEX). The differences in the respective instrument settings are the cause for different gMFI values within the same biosensor cell line as pointed out by the reviewer. We agree that ideally all experiments should be performed on the same instrument especially if further standardization as a diagnostic test is pursued. However, because of infrastructural restrictions (The flow cytometers at our center are heavily used by many researchers and it was especially hard to get time slots at the cytometers at the time when the experiments for this manuscript were performed) and the proof-of-concept nature of the experiments, especially during the initial time of this study, the use of different cytometers was unfortunately unavoidable. Importantly, all experiments that required data pooling from multiple single assays (such as measuring the serum donor cohort) were done on the same flow cytometer to ensure comparability. For each experiment the respective cytometer is now listed in the Source Data sheet for better transparency.

The question of this reviewer regarding the small molecule inhibitor prompted us to test several small molecules that have been proposed as inhibitor of PD-1/PD-L1 interaction in our assay. Four different compounds were tested and INCB and I3 showed excellent (INCB IC₅₀ 2.3 nM and moderate (I3 IC₅₀ 162nM) capacities to block PD1 and PDL1 interaction in our assay. By contrast

the other two compounds (BMS1 and BMS202) failed to block the engagement of PD1-zeta by PDL1 even high concentrations (10 μ M) and strongly impaired the viability of the reporter and stimulator cells when tested at higher concentrations. These results are summarized in Fig. S1e-k) of the revised manuscript. Of note unpublished results not included in the manuscript showed that BMS1 and BMS202 also failed to block the engagement of PD1 In a classical PD1 assay with PD1 reporter cells described previously by us (<https://doi.org/10.1038/s41598-019-47910-1>). Generally there is a lack of studies describing the assessment of small compound PD-1/PDL1 inhibitors in well-controlled reductionist cellular assays and we think that the data on (INCB; I3; BMS1 and BMS202) that are shown in the revised manuscript provide important information on the performance of small compound PD-1 blockers in a cellular assay which is much more informative about their potential than assays that measure their blocking capacity in a protein-based assay.

2. Figure 2, No signal attenuation is observed in Fig 2F. Utomilumab even promotes the signal intensity when treated with 0.1 μ g/ml. Please explain this result since Utomilumab is a humanized monoclonal antibody that activates 4-1BB while blocking binding to endogenous 4-1BBL, while Urelumab does not block the interaction of 4-1BB with its ligand (Blood,2018,131 (1): 49–57; Nat Commun. 2018,9(1):4679).

Response to reviewer 3

We are aware of these studies and have repeated these experiments using utomilumab (a fresh sample of the antibody was purchased for these experiments) at higher concentrations (up to 30 μ g/ml) and as an additional control we have used a polyclonal 4-1BB antibody preparation (AF838). Again, we did not find evidence for a ligand blocking capability of Utomilumab whereas the polyclonal 4-1BB antibody attenuated the signal indicating a weak blocking activity of this antibody preparation. The higher signal observed at 0.1 μ g/ml observed in the initially experiments was not seen indicating that they are likely to represent outliers. The results are shown in Figure 2g of the revised manuscript.

3. Figure7, when did the sera collect from donor 44C-46C (Fig. 7I-K)? Did they vaccinate with wild-type Spike? Why did individual serum samples show less neutralization capacity against wild-type SARS-CoV-2 than that of Omicron strain. This result is inconsistent with previous studies (N Engl J Med. 2022;387(1):86-88, PMID: 35731894; N Engl J Med. 2022,386(7):698-700, PMID: 35021005).

Response to reviewer 3

The sera were from donors vaccinated with wild-type Spike. We agree with this reviewer that the high neutralization capacity of these sera against the Omicron strain is surprising. Since our assay faithfully reproduced the reported data of the differential capacity of therapeutic antibodies in neutralizing spike proteins from different strains, we believe that the results are valid. The sera were collected in October 2022 and while the donors did not report testing positive since the appearance of Omicron, we cannot exclude that the donors have had contact with Omicron variants of SARS-CoV-2, especially because no regular PCR-testing of asymptomatic donors was performed. This study was not powered to detect the population wide immune escape by Omicron variants, however we could show that the IVIGs that had been purchased prior to the appearance of Omicron, showed a tendency to reduced neutralizing capacity against Omicron which would be in-line with the references stated by the reviewer.

4. The authors claim that the biosensor-based interaction assay is a highly accurate surrogate test for the detection of SARS-CoV-2 neutralizing antibodies. However, I am concerned about the data comparability (J Clin Virol. 2022,156: 105292. PMID: 36108404). The authors display

amount of data on neutralizing antibody titers that are not “normalized” and thus difficult to interpret without a reference cohort. In addition, the expression levels of the respective molecules (ACE2, Spike) vary among different cell clones. How to ensure the antibody titer comparison between different laboratories and over time?

Response to reviewer 3

We believe that our assay has the potential as to be used as surrogate assay for the detection of SARS-CoV-2 neutralizing antibodies. We fully agree with this reviewer that the implementation of our assay in different laboratories would require measures to guarantee data comparability. Therefore, as with any laboratory test, measures would have to be taken to control uniform and accurate performance of the assay in different laboratories. Such measures could e.g. be the use of defined sera, serum pools or blocking antibodies. Although the expression levels of the Spike proteins should have limited effects on the EC50 values, a careful matching of the expression levels of the Spike proteins would be mandatory for such an assay. We have toned down this claim in the manuscript to make it clear that additional work is required to establish our assay as surrogate test for the detection of SARS-CoV-2 neutralizing antibodies:

In the revised manuscript we write in the results section: “Altogether this indicates that the biosensor-based interaction assay has potential to be used as surrogate test for the detection of SARS-CoV-2 neutralizing antibodies.” and in the discussion “Our data indicate that this assay has the potential to be established as surrogate assay for the detection of SARS-CoV-2 neutralizing antibodies.”

5. Infection with SARS-CoV-2 is initiated by virus Spike binding to the ACE2, followed by fusion of the virus and cell membranes (Nature,2020,588,327–330; Sig Transduct Target Ther 2020,5,92, PMID: 32532959). Several peptides and antibodies recognize the conserved epitope in the S2 subunit of the Spike can inhibit SARS-CoV-2 infection by blocking the Spik-mediated membrane fusion (Microbiol Spectr. 2022,10(2): e0181421. PMID: 35293796; Science,2021,371(6536):1379-1382.). Can small molecules, peptides or anti-S2 antibodies block such interactions and reduce gMFI in the biosensor cell assay?

Response to reviewer 3

We have not performed experiments to test this but from our understanding our assay is unlikely to be affected by agents that interfere with membrane fusion.

6. As the authors point out, all their work is performed with a version of “wild-type” Spike protein without furin-cleavage site (RRRdel) and bearing the K1043P and V1043P mutations (Page 12 line 434). This could affect their data. Stimulator cells can be readily generated with full-length Spike protein, an analysis that should be performed.

Response to reviewer 3

For the revision we have performed experiments to show whether cells expressing unmutated spike proteins would also induced activation of the ACE2-biosensor cells. Unmutated spike protein was inducing reporter activation via engagement of ACE2-zeta and this activation was strongly reduced by the presence of a neutralizing serum. These experiments are shown in Figure S7 of the revised manuscript. We have performed extensive experiments for the revision of this manuscript and ask for understanding that we have not been able to perform a side by side comparison of mutated and unmutated spike proteins with different sera in our assay. However because of the concordance of the data obtained with our assay and the virus neutralization assay we think that it is unlikely that the mutation of the furin cleavage side will have a significant impact on the performance of the assay.

7. How did author define the limit of detection of biosensor interaction assay? What is the quantitative range of this assay for SARS-CoV-2 neutralizing antibody detection?

Response to reviewer 3:

Limit of detection: Sera that did not contain sufficient antibodies to significantly block the interaction of spike-expressing cells with ACE2- ζ reporter cells at a 1:3 fold dilution were considered not to contain relevant amounts of neutralizing antibodies. The quantitative range of the assay depends on the range of dilutions that are tested – usually the sera were tested in a range of 1:3 or 1:10 to 1:1000 (factor 3,16 dilution) or 1:10 to 1:5120 (factor 2 dilution). Sera with high neutralization capacity can be retested at higher dilutions thus increasing the range of the assay. We hope that we have understood this question correctly and have been able to provide a satisfactory answer.

Minor points:

1. For gene expression, please use the appropriate nomenclature (*italic*, etc.)

Response to reviewer 3:

These changes were made

2. Page 4, Table 2, why did the authors introduce K986P and V986P mutations in SARS-CoV-2-spike?

Response to reviewer 3:

These mutations were introduced since they were reported to improve the expression and stabilize the conformation of the SARS-CoV-2 spike protein. The K986P and V987P (in the initial manuscript we erroneously wrote V986P instead of V987P) mutations are included in many Covid-19 vaccines such as those from Moderna, Pfizer-BioNTech, Johnson & Johnson-Janssen, and Novavax (doi: [10.1038/s41467-023-37786-1](https://doi.org/10.1038/s41467-023-37786-1)).

3. Please provide a better resolution image. The font on Figures is too small, and the label needs to be clear.

Response to reviewer 3:

We have thoroughly revised the figures by improving their resolution and by increasing the font size of the labels that were too small.

4. Figure legends should be more informative for better understanding. Figure 1, please define BW and gMFI in the figure legend.

Response to reviewer 3:

The acronyms were defined in the figure legend.

5. Figure 1H, please provide the expression levels of CD80 and CD86 in biosensor cells. Is there a significant difference between CD80 and CD86 groups (Fig. 1H)?

Response to reviewer 3:

In the revised manuscript we show the expression of CD80 and CD86. Both molecules were strongly expressed on the K562 cells (Fig S2d). The expression was quantified using a QUANTUM-R-PE MESF kit. While CD80 is slightly higher expressed than CD86 there were no profound differences between the expression of these molecules (factor 1,25). We have compared the reporter gene expression induced by K562-CD80 and K562-CD86 using a Mann Whitney U test and the values were significantly higher in the K562-CD80 condition. We did not change the statistic in figure 1i since for us it make most to compare the each condition with the control.

6. Page 10 line 323: "As few as 2500 stimulator cells (reporter-to-stimulator ratio, 20:1)" In Fig 2E legend, 5×10^5 reporter cells were used, the ratio should be 200:1?

Response to reviewer 3:

We have used 5×10^4 reporter cells – this error was corrected in the revised manuscript – thank you for pointing this out to us.

7. The legends for Fig 2F and 2G are reversed.

Response to reviewer 3:

We have fixed this error in the revised manuscript.

8. Please define mKO2 in line 414.

Response to reviewer 3:

mKO2 (mKusabira-Orange2) was defined in the Methods section of the revised manuscript.

9. Figure 7, since the D614G was not observed in the Spike of Wuhan strain, did the author use different wild-type Spike proteins in 7A and 7B?

The designation "Wuhan" is not currently used anymore and was replaced by the designation "wild-type" in the revised version of the manuscript. As outlined in Table 1 the wild-type spike protein was adapted from the canonical sequence of the original SARS-CoV-2 virus (UniprotKB P0DTC2-1) and did not contain the D614G mutation. All later variants (B.1.617.2, AY.4.2, BA.1, BA.2 and BA.5) did contain the D614G mutation.

10. For SARS-CoV-2 neutralization assay, how much virus dose was used? How is a live virus titrated? Please also define the neutralization titer in detail and provide the NT50 values.

We have revisited and specified the description of the neutralization assay in the Methods section. A dose of TCID₅₀-100 of SARS-CoV-2 was preincubated with serially diluted serum and added to infection permissive Vero E6 cells. NT titers were determined as the highest reciprocal serum dilution at which no cytopathic effect was present. NT titers ≥ 10 were considered positive. NT50 values were not assessed in this study.

11. Could the author provide a workflow of the receptor-ligand interaction assay, including timing?

Response to reviewer 3:

The standard receptor ligand interaction assay really only involves the co-culture of the "biosensor" reporter cells with cells expressing the respective ligands (stimulator cells). In this study we have co-cultured 5×10^4 reporter and 2×10^4 stimulator cells for 24h as described in the method section.

To evaluate our system we have also analyzed the reporter activity at different time points (18-28h) and we used different numbers of reporter and stimulator cells and observed that neither the number, the ratio of the stimulator and reporter cells nor the time of the co-culture is a critical step in this assay.

Subsequently, the cells are analyzed by flow cytometry – Stimulator cells are excluded from the analysis either by their expression of RFP or by surface staining as described in the methods section. As for any cell-based assay it is important that it is performed under good cell culture practice conditions.

12. The receptor-ligand interaction assays in the method section requires more details. It will be useful to have more details on critical steps.

Response to reviewer 3:

We have taken care to provide detailed protocols for all the assays and experiments performed within this study. The receptor-ligand interaction assay itself really is very simple as pointed out in the answer to the previous points and there are no special or critical steps that have to be followed. We think that this is one of the advantages of this method.

REVIEWERS' COMMENTS

Reviewer #1 (Remarks to the Author):

I am satisfied that the authors have addressed the main points raised in the initial round of review and I'm pleased to recommend publication.

Reviewer #2 (Remarks to the Author):

The authors have satisfactorily addressed my suggestions.

Reviewer #3 (Remarks to the Author):

The authors have addressed most of my concerns and there is no more comments.